# DINOv2: Learning Robust Visual Features without Supervision

**Maxime Oquab**[**], **Timothée Darcet**[**], **Théo Moutakanni**[**],
**Huy V. Vo**[*], **Marc Szafraniec**[*], **Vasil Khalidov**[*], **Pierre Fernandez**, **Daniel Haziza**,
**Francisco Massa**, **Alaaeldin El-Nouby**, **Mahmoud Assran**, **Nicolas Ballas**, **Wojciech Galuba**,
**Russell Howes**, **Po-Yao Huang**, **Shang-Wen Li**, **Ishan Misra**, **Michael Rabbat**,
**Vasu Sharma**, **Gabriel Synnaeve**, **Hu Xu**, **Hervé Jegou**, **Julien Mairal**[1],
**Patrick Labatut**[*], **Armand Joulin**[*], **Piotr Bojanowski**[*]

*Meta AI Research*        [1]*Inria*

[*]core team        [**]equal contribution
Reviewed on OpenReview: `https://openreview.net/forum?id=a68SUt6zFt`

## Abstract

The recent breakthroughs in natural language processing for model pretraining on large quantities of data have opened the way for similar foundation models in computer vision. These models could greatly simplify the use of images in any system by producing general-purpose visual features, i.e., features that work across image distributions and tasks without finetuning. This work shows that existing pretraining methods, especially self-supervised methods, can produce such features if trained on enough curated data from diverse sources. We revisit existing approaches and combine different techniques to scale our pretraining in terms of data and model size. Most of the technical contributions aim at accelerating and stabilizing the training at scale. In terms of data, we propose an automatic pipeline to build a dedicated, diverse, and curated image dataset instead of uncurated data, as typically done in the self-supervised literature. In terms of models, we train a ViT model (Dosovitskiy et al., 2021) with 1B parameters and distill it into a series of smaller models that surpass the best available general-purpose features, OpenCLIP (Ilharco et al., 2021) on most of the benchmarks at image and pixel levels.

## 1 Introduction

Learning task-agnostic pretrained representations have become the standard in Natural Language Processing (NLP) (Radford et al., 2019; Raffel et al., 2020; Chowdhery et al., 2022; Hoffmann et al., 2022; Touvron et al., 2023). One can use these features "as they are", i.e., without fine-tuning, and achieve performances on downstream tasks that are significantly better than those produced by task-specific models (Brown et al., 2020). This success has been fueled by pretraining on large quantities of raw text using pretext objectives, such as language modeling (Radford et al., 2017) or word vectors (Devlin et al., 2019), that require no supervision.

Following this paradigm shift in NLP, we expect similar "foundation" models to appear in computer vision (Bommasani et al., 2021). These models should generate visual features that work out of the box on any task, both at the image level, e.g., image classification, and pixel level, e.g., segmentation. Most promising efforts towards these foundation models focus on text-guided pretraining, i.e., using a form of textual supervision to guide the training of the features (Joulin et al., 2016; Mahajan et al., 2018; Radford et al.,

---

All the authors are affiliated to Meta, except Julien Mairal who is affiliated to Inria. Timothée Darcet and Pierre Fernandez have a co-affiliation with Inria. Théo Moutakanni has a co-affiliation with Université Paris Saclay. Alaaeldin El-Nouby has a co-affiliation with Inria and ENS-PSL. Correspondence: {qas, timdarcet, theomoutakanni, ajoulin, bojanowski}@meta.com

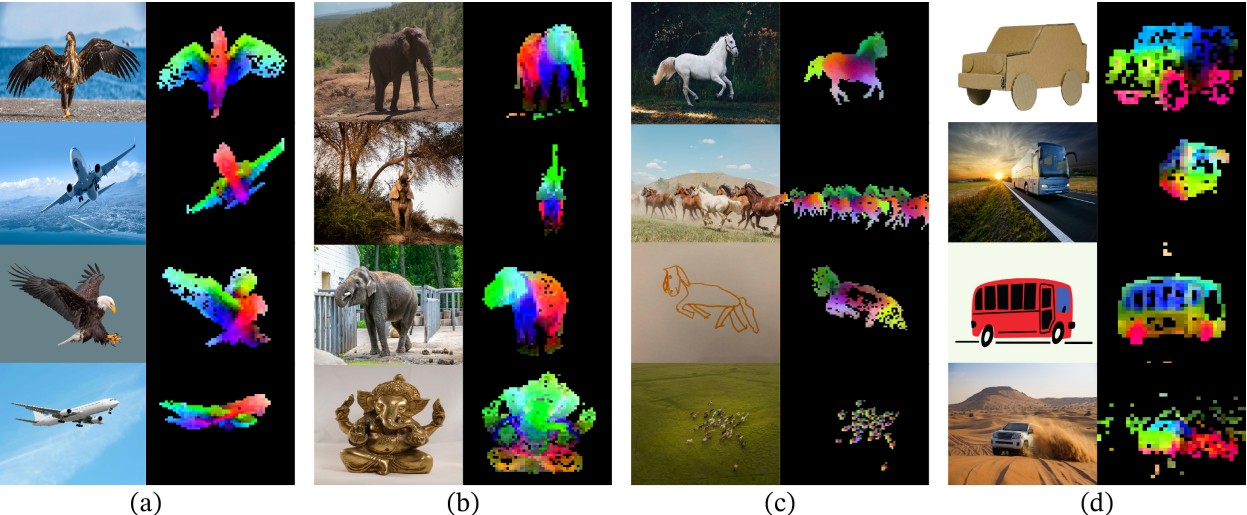

(a) (b) (c) (d)

Figure 1: **Visualization of the first PCA components.** We compute a PCA between the patches of the images from the same column (a, b, c and d) and show their first 3 components. Each component is matched to a different color channel. Same parts are matched between related images despite changes of pose, style or even objects. Background is removed by thresholding the first PCA component.

2021). This form of text-guided pretraining limits the information that can be retained about the image since captions only approximate the rich information in images, and complex pixel-level information may not surface with this supervision. Furthermore, these image encoders require aligned text-image corpora and hence, do not offer the flexibility of their text counterparts, that is, to learn from raw data alone.

An alternative to text-guided pretraining is self-supervised learning (Caron et al., 2018; Chen et al., 2020; He et al., 2022) where features are learned from images alone. These approaches are conceptually closer to pretext tasks such as language modeling and can capture information at the image and pixel level (Caron et al., 2021). Additionally, the features output by self-supervised models have been shown to exhibit various useful properties, and have enabled enabled a wide variety of applications (Amir et al., 2022; Tumanyan et al., 2022; Ofri-Amar et al., 2023; Hamilton et al., 2022). However, despite their potential to learn general-purpose features, most of the advances in self-supervised learning were made in the context of pretraining on a small curated dataset, ImageNet-1k (Russakovsky et al., 2015). Some efforts on scaling these approaches beyond ImageNet-1k have been attempted (Caron et al., 2019; Goyal et al., 2021; 2022a), but they focused on uncurated datasets, which typically lead to a significant drop in the quality of the features. This is explained by the lack of control over the data quality and diversity, which are essential to produce good features.

In this work, we explore if self-supervised learning has the potential to learn general-purpose visual features if pretrained on a large quantity of curated data. We revisit existing discriminative self-supervised approaches that learn features at both the image and patch level, such as iBOT (Zhou et al., 2022a), and we reconsider some of their design choices under the lens of a larger dataset. Most of our technical contributions are tailored toward stabilizing and accelerating discriminative self-supervised learning when scaling in model and data sizes. These improvements make our approach around $2\times$ faster and require $3\times$ less memory than similar discriminative self-supervised methods, allowing us to leverage longer training with larger batch sizes.

Regarding pretraining data, we have built an automatic pipeline to filter and rebalance datasets from an extensive collection of uncurated images. This pipeline is inspired by pipelines used in NLP (Wenzek et al., 2020), where data similarities are used instead of external metadata and do not require manual annotation. A major difficulty when dealing with images in the wild is to rebalance concepts and avoid overfitting on a few dominant modes. In this work, a naive clustering approach works reasonably well to resolve this issue. We gathered a small but diverse corpus of 142M images to validate our approach.

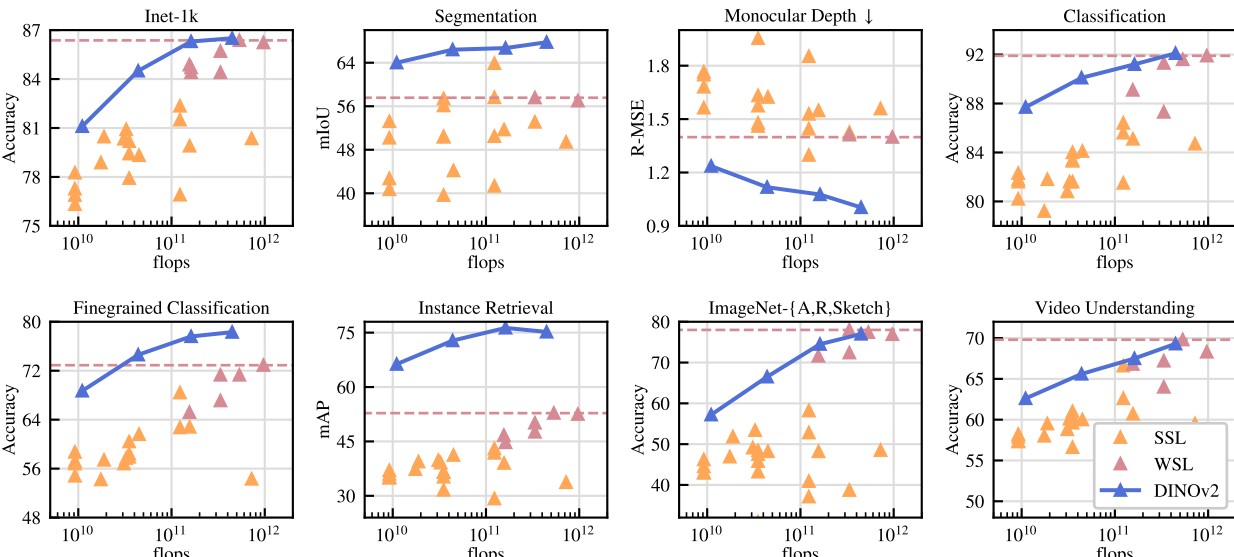

Figure 2: **Evolution of performance when scaling in parameters.** We show performance on eight types of vision tasks, as presented in Sec. 7, and average metrics with each type. Features are extracted from our self-supervised encoders, DINOv2 (dark blue), and we compare them with self-supervised methods (pale orange), as well as weakly-supervised methods (dark pink). We report the best-performing weakly-supervised model's performance as a dashed horizontal line. Our family of models drastically improves over the previous state of the art in self-supervised learning and reaches performance comparable with weakly-supervised features. See Sec. 7 for a detailed analysis.

Finally, we provide a variety of pretrained visual models, called DINOv2, trained with different Vision Transformers (ViT) (Dosovitskiy et al., 2016) architectures on our data. We release all the models and the code to retrain DINOv2 on any data. We validate the quality of DINOv2 on various computer vision benchmarks at both image and pixel levels as we scale them, as summarized in Fig. 2. We conclude that self-supervised pretraining alone is a good candidate for learning transferable frozen features that are competitive with the best openly available weakly-supervised models.

## 2 Related Work

**Intra-image self-supervised training.** A first family of self-supervised methods focuses on pretext tasks built from the image, i.e., extracting a signal from the image to be predicted from the rest of the image. This idea has become prevalent with the work of Doersch et al. (2015), where they train by predicting the context of a given patch. Many other pretext tasks were introduced based on, for example, re-colorizing images (Zhang et al., 2016), predicting transformations (Gidaris et al., 2018), inpainting (Pathak et al., 2016) or patch re-ordering (Noroozi & Favaro, 2016; Misra & Maaten, 2020). Recently, the emergence of patch-based architectures, like ViTs, has led to a revisit of inpainting for pre-training (He et al., 2022; Bao et al., 2021; El-Nouby et al., 2021), potentially in feature space (Assran et al., 2023; Baevski et al., 2022). Of particular interest, He et al. (2022) show that a masked auto-encoder (MAE) learns features that provide substantial improvements when finetuned on downstream tasks. This property of MAEs has been further validated on video (Tong et al., 2022), audio (Xu et al., 2022), and across other modalities (Girdhar et al., 2023). However, their features require supervised finetuning, while our features perform well out of the box.

**Discriminative self-supervised learning.** The second line of work, closer to ours, is using discriminative signals between images or groups of images to learn features. This family of methods has roots in early deep learning work (Hadsell et al., 2006) but became popular with the emergence of instance classification methods (Dosovitskiy et al., 2016; Bojanowski & Joulin, 2017; Wu et al., 2018). Several improvements

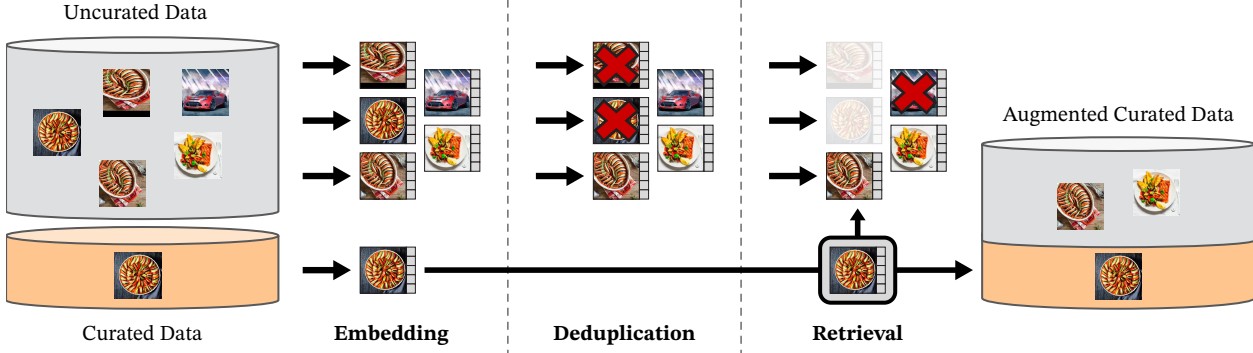

Figure 3: **Overview of our data processing pipeline.** Images from curated and uncurated data sources are first mapped to embeddings. Uncurated images are then deduplicated before being matched to curated images. The resulting combination augments the initial dataset through a self-supervised retrieval system.

were made based either on instance-level objectives (Hénaff et al., 2019; He et al., 2020; Chen & He, 2021; Chen et al., 2020; Grill et al., 2020; Caron et al., 2021) or clustering (Caron et al., 2018; Asano et al., 2020; Caron et al., 2020). These methods provide performant frozen features on standard benchmarks like ImageNet (Russakovsky et al., 2015), but they are hard to scale to larger model sizes (Chen et al., 2021). In this work, we revisit the training of these approaches in the context of large pretraining datasets and models. In particular, we build on top of Zhou et al. (2022a) that we find particularly suited for scaling.

**Scaling self-supervised pretraining.** A growing body of work has focused on the scaling abilities of self-supervised learning in terms of data and model size (Caron et al., 2019; Goyal et al., 2019; Tian et al., 2021; Goyal et al., 2022a). Most of these works use large quantities of uncurated data to train models without supervision. They show evidence that discriminative methods scale with data, but because of the poor quality of the pretraining data, most of the results are obtained by finetuning the features. Of particular interest, Goyal et al. (2021) have also shown that these methods benefit from scaling in model size given enough pretrained data. This line of work questions the ability of self-supervised methods to work on any data while we focus on producing the best pretrained encoders.

**Automatic data curation.** Our dataset construction borrows from the image retrieval community (Weinzaepfel et al., 2021; Radenović et al., 2018b; Berman et al., 2019; Douze et al., 2009; Tolias et al., 2016; Revaud et al., 2019). In particular, the use of retrieval to augment the training set has been studied in the context of semi-supervised learning (Yalniz et al., 2019). Similarly, others have used hashtags or other metadata (Mahajan et al., 2018; Radford et al., 2021) or pretrained vision encoders (Schuhmann et al., 2021; 2022) to filter uncurated datasets. Unlike these works, we use no pretrained encoders, metadata nor supervision to filter images and leverage visual similarity between images. Our approach is inspired by text curation pipelines (Wenzek et al., 2020), where a language model is trained on Wikipedia to score texts extracted from an uncurated source.

## 3 Data Processing

We assemble our curated LVD-142M dataset by retrieving, from a large pool of uncurated data, images that are close to those in several curated datasets. We describe below the main components in our data pipeline including the curated/uncurated data sources, the image deduplication step and the retrieval system. Our pipeline does not require any metadata or text and directly works with images, as shown in Fig. 3. We refer the reader to appendix A for more details on our approach.

**Data sources.** Our selection of curated datasets is detailed in the appendix (Table 15) and contains ImageNet-22k, the train split of ImageNet-1k, Google Landmarks and several fine-grained datasets. For the

uncurated data source, we collect a raw unfiltered dataset of images from a publicly available repository of crawled web data. From each web page in the repository, we extract URL links of images from `` tags. We discard URLs that are unsafe or restricted by domains, and post-process the downloaded images (PCA hash deduplication, NSFW filtering, and blurring identifiable faces). This results in 1.2B unique images.

**Deduplication.** We apply the copy detection pipeline of Pizzi et al. (2022) to the uncurated data and remove near-duplicate images. This reduces redundancy and increases diversity among images. We also remove near-duplicates of images contained in the test or validation set of any benchmark used in this work.

**Self-supervised image retrieval.** We build our curated pretraining dataset by retrieving images from our uncurated data source that are close to images in our curated sources. In order to do this, we first compute an image embedding using a self-supervised ViT-H/16 network pretrained on ImageNet-22k, and use cosine-similarity as a distance measure between images. Then, we perform k-means clustering of the uncurated data. Given a query dataset for retrieval, if it is large enough we retrieve $N$ (typically 4) nearest neighbors for each query image. If it is small, we sample $M$ images from the cluster corresponding to each query image. Although visual inspection seemed to indicate good retrieval quality for $N$ much larger than 4, this leads to more collisions (images that are nearest-neighbor retrievals of multiple queries). We choose $N = 4$ as it provides a good tradeoff in that sense.

**Implementation Details.** The deduplication and retrieval stages of our pipeline rely on the Faiss library (Johnson et al., 2019) to efficiently index and compute batch searches of nearest embeddings. In particular, we heavily leverage its support for GPU-accelerated indices, using inverted file indices with product quantization codes (Jegou et al., 2010). The whole processing is distributed on a compute cluster of 20 nodes equipped with 8 V100-32GB GPUs and takes less than two days to produce the LVD-142M dataset.

## 4 Discriminative Self-supervised Pre-training

We learn our features with a discriminative self-supervised method that can be seen as a combination of DINO and iBOT losses with the centering of SwAV (Caron et al., 2020). We also add a regularizer to spread features and a short high-resolution training phase. We rapidly introduce each of these approaches, but more details can be found in the related papers, or in our open-sourced code.

- **Image-level objective (Caron et al., 2021).** We consider the cross-entropy loss between the features extracted from a student and a teacher network. Both features are coming from the class token of a ViT, obtained from different crops of the same image. We pass the student class token through the student DINO head. This head is an MLP model outputting a vector of scores, that we call "prototype scores". We then apply a softmax to obtain $p_s$. Similarly, we apply the teacher DINO head to the teacher class token to obtain teacher prototype scores. We then apply a softmax followed by a centering with moving average (or a Sinkhorn-Knopp centering as detailed thereafter) to obtain $p_t$. The DINO loss term corresponds to:

$$\mathcal{L}_{DINO} = -\sum p_t \log p_s$$

  We learn the parameters of the student and build the teacher head with an exponential moving average of past iterates (He et al., 2020).

- **Patch-level objective (Zhou et al., 2022a).** We randomly mask some of the input patches given to the student, but not to the teacher. We then apply the student iBOT head to the student mask tokens. Similarly, we apply the teacher iBOT head to the (visible) teacher patch tokens corresponding to the ones masked in the student. We then apply the softmax and centering steps as above, and obtain the iBOT loss term:

$$\mathcal{L}_{iBOT} = -\sum_i p_{ti} \log p_{si}$$

, where $i$ are patch indices for masked tokens. Similarly to above, we learn the parameters of the student, and build the teacher head through exponential moving average.

- **Untying head weights between both objectives.** Both the DINO and the iBOT loss use a learnable MLP projection head. It is applied to the output tokens and the loss is compute atop. In Zhou et al. (2022a), an ablation study shows that sharing parameters between the DINO and iBOT heads leads to better performance. At scale, we observed that the opposite is true, and we therefore use two separate heads in all our experiments.

- **Sinkhorn-Knopp centering (Caron et al., 2020).** Ruan et al. (2023) recommend to replace the teacher softmax-centering step of DINO and iBot by the Sinkhorn-Knopp (SK) batch normalization of SwAV (Caron et al., 2020). We run the Sinkhorn-Knopp algorithm steps for 3 iterations. For the student, we apply the softmax normalization.

- **KoLeo regularizer (Sablayrolles et al., 2019).** The KoLeo regularizer derives from the Kozachenko-Leonenko differential entropy estimator (see Beirlant et al. (1997); Delattre & Fournier (2017)) and encourages a uniform span of the features within a batch. Given a set of $n$ vectors $(x_1, \ldots, x_n)$, it is defined as

$$\mathcal{L}_{\text{koleo}} = -\frac{1}{n} \sum_{i=1}^{n} \log(d_{n,i}),$$

where $d_{n,i} = \min_{j \neq i} \|x_i - x_j\|$ is the minimum distance between $x_i$ and any other point within the batch. We also $\ell_2$-normalize the features before computing this regularizer.

- **Adapting the resolution (Touvron et al., 2019).** Increasing image resolution is key to pixel-level downstream tasks such as segmentation or detection, where small objects disappear at low resolutions. However, training at high resolution is time and memory demanding, and instead, we increase the resolution of images to $518 \times 518$ during a short period at the end of pretraining. This is also similar to UniViT training from Likhomanenko et al. (2021) and FlexiViT training from Beyer et al. (2023).

## 5 Efficient implementation

We consider several improvements to train models at a larger scale. We train models on A100 GPUs using PyTorch 2.0. The code and pretrained models are made available under Apache 2.0 license [1]. The details of our models are in the appendix, Table 17. With the same hardware, compared to the iBOT implementation, the DINOv2 code runs around 2× faster using only 1/3 of the memory.

**Fast and memory-efficient attention.** We implemented our own version of FlashAttention (Dao et al., 2022) to improve memory usage and speed on the self-attention layers. Our version is on par with or better than the original on all cases considered, while covering more use-cases and hardware. Due to the GPU hardware specifics, the efficiency is best when the embedding dimension per head is a multiple of 64, and the matrix operations are even better when the full embedding dimension is a multiple of 256. As a consequence, our ViT-g architecture slightly differs from the architecture proposed by Zhai et al. (2022) in order to maximize compute efficiency, and we use an embedding dimension of 1536 with 24 heads (64 dim/head), rather than 1408 with 16 heads (88 dim/head). Our experiments did not show significant differences in final accuracy, and our ViT-g backbone counts 1.1B parameters.

**Sequence packing.** The DINO algorithm requires forwarding both large crops (at resolution 224) and small crops (resolution 98). When split into patches, these two groups are represented by token sequences of different lengths and cannot be forwarded together. In order to accelerate training, we use a trick called "sequence packing," which originates from NLP (Krell et al., 2022). The idea is simple: we concatenate the

---

[1] https://github.com/facebookresearch/dinov2

sequences we must forward through the transformers into a single long sequence. We pass this sequence through the transformer blocks as usual. However, a block-diagonal mask is applied to the self-attention matrix in attention layers, preventing attention between different sequences. This way, the forward is strictly equivalent to forwarding each sequence separately. This trick gives us significant compute efficiency gains compared to using separate forward and backward passes, as in prior implementations. The lower-level components of our setup are available in the xFormers library[2] (Lefaudeux et al. (2022)).

**Efficient stochastic depth.** We implement an improved version of stochastic depth (Huang et al., 2016) that skips the computation of the dropped residuals rather than masking the result. This saves memory and compute in proportion approximately equal to the drop rate, thanks to specific fused kernels. With high drop rates ($d = 40\%$ in this work), this allows a drastic improvement in compute efficiency and memory usage. The implementation consists of randomly shuffling the $B$ samples over the batch dimension, and slicing the first $(1 - d) \times B$ samples for the computations in the block.

**Fully-Sharded Data Parallel (FSDP).** Minimizing our objective with the AdamW optimizer requires 4 model replicas in float32 precision – student, teacher, optimizer first moments, optimizer second moments. This sums to 16 GB of memory for a billion-parameter model such as our ViT-g. In order to reduce this memory footprint per GPU, we split the model replicas across GPUs, i.e., sharding 16 GB across GPUs using the PyTorch implementation of FSDP. Consequently, the model size is not bounded by the memory of a single GPU but by the total sum of GPU memory across compute nodes. The Pytorch implementation of FSDP brings a second advantage, which is to save on the cross-GPU communication costs: the weight shards are stored in float32 precision as required by the optimizer, but broadcasting weights and reducing gradients is done in float16 precision for the backbone (MLP heads gradients are reduced in float32 to avoid training instabilities). This leads to approximately 50% reduction in communication costs compared to the float32 gradient all-reduce operation used in DistributedDataParallel (DDP), which is used in other self-supervised pretraining methods (Caron et al., 2021; Zhou et al., 2022a). As a consequence, the training procedure scales more efficiently than DDP with float16 autocast when scaling the number of GPU nodes. Overall, Pytorch-FSDP mixed-precision is superior to DDP with autocast in virtually all cases we encountered.

**Model distillation.** Most of our technical improvements to the training loop aim at improving the training of large models over large quantities of data. For smaller models, we distill them from our largest model, the ViT-g, instead of training them from scratch. Knowledge distillation (Hinton et al., 2014) aims at reproducing the output of a large model with a smaller model by minimizing some distance between both outputs for a set of given inputs. Since our objective function is a form of distillation from the teacher network to the student network, we leverage the same training loop with a few exceptions: we use a larger model as a frozen teacher, keep a spare EMA of the student that we use as our final model, remove the masking and stochastic depth, and, apply the iBOT loss on the two global crops. In our ablations, we observe that this approach achieves better performance than training from scratch, even for a ViT-L. Our distillation method ends up close to the one described by Duval et al. (2023), except we do not modify the loss terms for distillation and evaluate the EMA of the student.

## 6 Ablation Studies

We present a set of ablations to empirically validate different components of our pipeline: the technical modifications described in Sec. 4, the pretraining data and the impact of model distillation. We consider various downstream tasks that are described in Sec. 7.

### 6.1 Improved Training Recipe

Our approach improves over the iBOT method by combining it with several existing components described in Sec. 4. To evaluate their importance, we train multiple models where we successively add components to a baseline iBOT model. We report the Top-1 accuracy on the validation set of ImageNet-1k with a k-NN

---

[2]https://github.com/facebookresearch/xformers

|  | INet-1k k-NN | INet-1k linear |
|---|---|---|
| iBOT | 72.9 | 82.3 |
| +(our reproduction) | 74.5 ↑1.6 | 83.2 ↑0.9 |
| +LayerScale, Stochastic Depth | 75.4 ↑0.9 | 82.0 ↓1.2 |
| +128k prototypes | 76.6 ↑1.2 | 81.9 ↓0.1 |
| +KoLeo | 78.9 ↑2.3 | 82.5 ↑0.6 |
| +SwiGLU FFN | 78.7 ↓0.2 | 83.1 ↑0.6 |
| +Patch size 14 | 78.9 ↑0.2 | 83.5 ↑0.4 |
| +Teacher momentum 0.994 | 79.4 ↑0.5 | 83.6 ↑0.1 |
| +Tweak warmup schedules | 80.5 ↑1.1 | 83.8 ↑0.2 |
| +Batch size 3k | 81.7 ↑1.2 | 84.7 ↑0.9 |
| +Sinkhorn-Knopp | 81.7 = | 84.7 = |
| +Untying heads = DINOv2 | 82.0 ↑0.3 | 84.5 ↓0.2 |

Table 1: **Ablation study of the training differences between iBOT and DINOv2.** We optimize for k-NN performance, as in our experience, the linear probe performance is lower-bounded by the k-NN performance. Some modifications, like LayerScale and a high Stochastic Depth (rate=0.4), incur a decrease in linear probe performance, but have the benefits of increasing the stability of training by avoiding NaN loss values during training (Touvron et al., 2022). Overall, these modifications allowed for the next set of improvements to be added. Experiments are run using the ViT-Large architecture on ImageNet-22k.

| Training Data | INet-1k | Im-A | ADE-20k | Oxford-M | iNat2018 | iNat2021 | Places205 |
|---|---|---|---|---|---|---|---|
| INet-22k | **85.9** | 73.5 | 46.6 | 62.5 | 81.1 | 85.6 | 67.0 |
| INet-22k \ INet-1k | 85.3 | 70.3 | 46.2 | 58.7 | 80.1 | 85.1 | 66.5 |
| Uncurated data | 83.3 | 59.4 | 48.5 | 54.3 | 68.0 | 76.4 | 67.2 |
| LVD-142M | 85.8 | **73.9** | **47.7** | **64.6** | **82.3** | **86.4** | **67.6** |

Table 2: **Ablation of the source of pretraining data.** We compare the INet-22k dataset that was used in iBOT to our dataset, LVD-142M. Each model is trained for the same number of iterations, that is smaller than in our final run, without high-resolution adaptation. Pretraining on LVD-142M maintains the performance over INet-1k while leading to models that perform better in other domains.

and a linear probe in Table 1. Generally, we observe that each component improves the performance on either k-NN or linear probing and even both in most cases. Only LayerScale and Stochastic Depth incur a performance drop in linear probing but significantly improve the training stability in our experience.

## 6.2 Pretraining Data Source

The quality of features is directly related to the quality of the pretraining data. In this experiment, we probe the impact of LVD-142M compared to ImageNet-22k, a commonly used pretraining dataset, or using directly raw and uncurated data. For the uncurated dataset, we randomly sample 142 million images from the same data source as LVD-142M. We train a ViT-g/14 on each dataset for the same number of iterations. We also include a variant of ImageNet-22k obtained by removing the synsets of ImageNet-1k (INet-22k \ INet-1k) for completeness. We report the comparisons in Table 2.

The most salient observation is that training on a curated set of images works better on most benchmarks than training on uncurated data. This confirms the benefit of curating data, even in the case of self-supervised pretraining. When compared with models trained on ImageNet-22k, training on LVD-142M is also superior on all the benchmarks but ImageNet-1k. This confirms that training on a more diverse set of images improves the quality of the features in domains that are not covered by ImageNet-22k. We also see that training on our curated data increases the performances on domains that are not used for the curation process (INaturalist 2018, 2021 and Places205), proving that scale and diversity can benefit unseen domains.

Figure 4: **Model scale versus data scale.** Evolution of performance as a function of model size for two different pretraining datasets: ImageNet-22k (14M images) and LVD-142M (142M images). The ViT-g trained on LVD-142M surpasses the ViT-g trained on ImageNet-22k on most benchmarks.

| KoLeo | INet-1k | Im-A | ADE-20k | Oxford-M | | MIM | INet-1k | Im-A | ADE-20k | Oxford-M |
|---|---|---|---|---|---|---|---|---|---|---|
| ✗ | 85.3 | 70.6 | 47.2 | 55.6 | | ✗ | 85.3 | 72.0 | 44.2 | 64.3 |
| ✓ | 85.8 | 72.8 | 47.1 | 63.9 | | ✓ | 85.8 | 72.8 | 47.1 | 63.9 |

(a) Koleo loss                         (b) MIM objective in iBOT

Table 3: **(a)** Effect of the KoLeo loss term. **(b)** Effect of the iBOT Masked Image Modeling (MIM) loss term. Evaluation performed on ImageNet-{1k,A} (classification with linear probe, accuracy %), ADE-20k (segmentation with linear layer, mIoU) and Oxford-M (image retrieval, mAP). Each model is trained on the same number of iterations, that is smaller than our final run. The KoLeo loss term improves nearest-neighbor search tasks (e.g. retrieval), and the MIM loss improves patch-level tasks (e.g. segmentation).

Overall, the conclusion of this ablation is that our dataset provides a good balance of different types of images that leads to the best performance overall.

## 6.3   Model Size and Data

We quantify the importance of scaling data with the model size in Fig. 4. As the size of models grow, training on LVD-142M becomes more beneficial than training on ImageNet-22k. For instance, a ViT-g trained on LVD-142M matches the performance on ImageNet-1k of a model trained on ImageNet-22k while significantly outperforming it on the other benchmarks.

## 6.4   Loss Components

We validated the proposed technical improvements in Sec. 6.1 by adding them incrementally. This section analyzes the performance hit observed if we ablate specific loss terms, starting from our best-performing model. We ablate the importance of the KoLeo loss and the impact of the masked image modeling term. For both, we report performance on ImageNet-1k using a linear classifier, ADE-20k segmentation using a linear classifier, and nearest-neighbor image retrieval on Oxford-M. Table 3a shows the impact of using the KoLeo loss. We see that the instance retrieval performance improves by more than 8%, confirming that this term helps spread features in the output space. At the same time, the other metrics do not suffer from this regularization. In Table 3b, we show the impact of using the masked image modeling term from iBOT. This term is critical for dense prediction tasks, leading to almost 3% performance improvement.

## 6.5   Impact of Knowledge Distillation

For small architectures, we distill larger models instead of training them from scratch. We use the distillation procedure described in Sec. 5. We evaluate the effectiveness of this approach by comparing a ViT-L/14 trained from scratch with one distilled from a ViT-g/14 over 12 benchmarks in Fig. 5. We also report the performance of the ViT-g/14 used for distillation as a topline. The distilled model outperforms the one trained from scratch on all 12 benchmarks, validating our pretraining approach for small models.

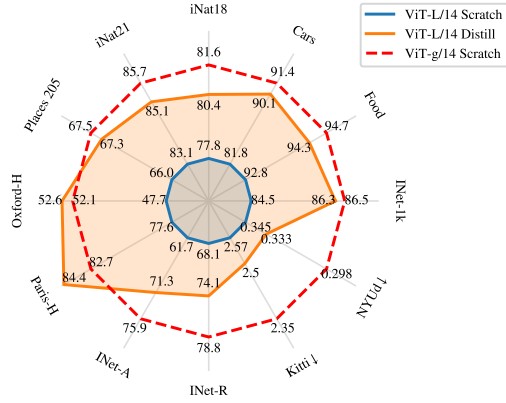

| Arch | Method | INet-1k | Segm. | Depth↓ | Classif. |
|---|---|---|---|---|---|
| ViT-g/14 | Scratch | 86.5 | 73.4 | 1.00 | 92.1 |
| ViT-L/14 | Scratch | 84.5 | 72.2 | 1.10 | 90.2 |
| ViT-L/14 | Distill | **86.3** | **73.3** | **1.08** | **91.2** |

| Arch | Method | Finegr. | Retriev. | ARSketch | Video |
|---|---|---|---|---|---|
| ViT-g/14 | Scratch | 78.3 | 75.2 | 77.0 | 69.3 |
| ViT-L/14 | Scratch | 75.8 | 71.3 | 69.5 | 67.3 |
| ViT-L/14 | Distill | **77.6** | **76.3** | **74.5** | **67.5** |

(a) Comparison on individual metrics          (b) Averaged metrics on 8 vision tasks

Figure 5: **Effectiveness of knowledge distillation.** Comparison between a ViT-L trained from scratch or distilled from DINOv2 using ViT-g/14. For reference, we also report the performance of the ViT-g/14 teacher. We show that a ViT-L model distilled from a frozen ViT-g outperforms a the same model trained from scratch on all benchmarks, sometimes even outperforming the distillation target.

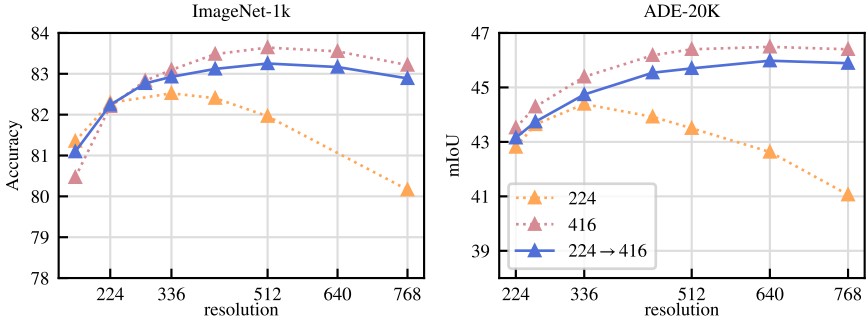

Figure 6: **Role of resolution.** Performance of ViT-L/16 trained on ImageNet-1k at fixed resolution ("224" and "416") or trained at 224 then 416 for a short duration ("224→416"). We train linear classifiers on top of frozen features at different resolutions and report Top-1 accuracy on ImageNet and mIoU on ADE-20k. We observe that performing SSL training at high resolution for a short duration achieve behavior and results close to training at the same high resolution for the full training, at a fraction of the cost.

## 6.6 Impact of Resolution

We measure the impact of changing the resolution during the pretraining on the performance of image and patch-level features. We consider models trained from scratch using a fixed resolution of either $224 \times 224$ or $416 \times 416$, and a model trained from scratch at $224 \times 224$, then resumed for 10k more iterations at $416 \times 416$. High-resolution training is compute-intensive, so we conduct this ablation on a small setup: a ViT-L/16 trained on ImageNet1k. In Fig. 6, we report the performance of a linear probe on ImageNet-1k and ADE-20k, evaluated at various resolutions. The model trained on high-resolution images performs best across resolutions, but this comes at a high cost: training at 416 is approximately $3 \times$ more compute-intensive than training at 224. On the other hand, training at high resolution for only 10k iterations at the end of the training is almost as good and only requiring a fraction of the compute. As a consequence, we include this step at the end of the training rather than training at a high resolution from scratch.

# 7   Results

In this section, we present the empirical evaluation of our models on many image understanding tasks. We evaluate both global and local image representations, on category and instance-level recognition, semantic segmentation, monocular depth prediction, and action recognition. We detail the list of benchmarks in Appendix C. The goal of this evaluation is twofold. First, we show that our self-supervised features outperform the current state of the art by a very large margin. Second, we show that they match, or surpass the performance of weakly-supervised ones on a substantial number of tasks.

**Baselines.** In our comparisons, we use two kinds of models as baselines. We compare to the best performing self-supervised models that are openly available. First, we run our evaluations for MAE (He et al., 2022), DINO (Caron et al., 2021), SEERv2 (Goyal et al., 2022a), MSN (Assran et al., 2022), EsViT (Li et al., 2022a), Mugs (Zhou et al., 2022b) and iBOT (Zhou et al., 2022a). When several architectural variants were proposed for a given method, we report results for the one that leads to best top-1 accuracy on ImageNet-1k. Second, we report performance of open-source weakly-supervised models such as CLIP (Radford et al., 2021), OpenCLIP (Ilharco et al., 2021; Cherti et al., 2023), and SWAG (Singh et al., 2022). When evaluating models on ImageNet-1k, we report the performance for each of the aforementioned methods. For all other evaluations, we report the four best-performing models amongst SSL ones. Also, for reference, we report the best performing OpenCLIP-G for weakly-supervised ones.

## 7.1   ImageNet Classification

As a first evaluation, we probe the quality of the holistic image representation produced by the model on the ImageNet-1k classification dataset. We evaluate the quality of features by training a simple classifier over a frozen backbone, and do not perform finetuning of the backbone weights. Following previous work, we use a linear model for simplicity, ensuring a reproducible evaluation, despite the fact that classes may not be linearly separable. Because most SSL methods were developed using ImageNet-1k validation performance as a debugging signal, we also report the top-1 accuracy on ImageNet-ReaL and ImageNet-V2. In order to report this additional validation performance, for all models, we run the evaluation with our code. We compare our frozen features to the best publicly available SSL features in Table 4, regardless of architecture or pretraining data. We see the components proposed in this work lead to a very significant improvement (+4.2%) over the previous state of the art (iBOT ViT-L/16 trained on ImageNet-22k) on linear evaluation. At the same time, we also see that the performance increase on the alternative test sets is larger for our method, indicating stronger generalization. We describe details of our linear evaluation in Appendix B.3.

**How far are we from weakly-supervised models?** We also want to validate that our features are competitive with state-of-the-art open-source weakly supervised models. To this end, we compare on ImageNet-1k, using the linear evaluation, to three off-the-shelf methods with several architectural variants. For all models, we run the linear evaluation using our code, after making sure that our numbers match those reported in technical reports and papers. We show the result of this evaluation in Table 4. We see that our backbone, surpases the performance of OpenCLIP with a ViT-G/14 architecture (+0.3%) and EVA-CLIP with a ViT-g/14 (+0.1%). At the same time, we also observe that our performance on the ImageNet-V2 test set is significantly better (+1.1% versus EVA-CLIP), indicating better generalization. For the remainder of this section, we report OpenCLIP-G as a reference for weakly-supervised models.

**Can we finetune the encoders?** We question if the ability of our models to produce high quality frozen features impact their performance when finetuned with supervision on a specific dataset. While this is not core to this paper, this experiment is indicative of whether we have involuntarily specialized our models to the setting of linear evaluations of frozen features. To run this sanity check, we apply the finetuning pipeline from Touvron et al. (2022), without tweaking hyper-parameters. In Table 5, we show that the Top-1 accuracy on the validation set of ImageNet-1k improves by more than +2% when the backbone is finetuned. This is true both when using models at resolution 224 and 448. Further gains can be obtained by tuning the hyper-parameters of the finetuning, but this is beyond the goal of this sanity check. Nonetheless, our best finetuned performance (88.9%) is only a couple of percent below (−2.2%) the absolute state of the

| Method | Arch. | Data | Text sup. | kNN | linear | | |
|--------|-------|------|-----------|-----|--------|---|---|
| | | | | val | val | ReaL | V2 |
| **Weakly supervised** | | | | | | | |
| CLIP | ViT-L/14 | WIT-400M | ✓ | 79.8 | 84.3 | 88.1 | 75.3 |
| CLIP | ViT-L/14$_{336}$ | WIT-400M | ✓ | 80.5 | 85.3 | 88.8 | 75.8 |
| SWAG | ViT-H/14 | IG3.6B | ✓ | 82.6 | 85.7 | 88.7 | 77.6 |
| OpenCLIP | ViT-H/14 | LAION-2B | ✓ | 81.7 | 84.4 | 88.4 | 75.5 |
| OpenCLIP | ViT-G/14 | LAION-2B | ✓ | 83.2 | 86.2 | 89.4 | 77.2 |
| EVA-CLIP | ViT-g/14 | custom* | ✓ | **83.5** | 86.4 | 89.3 | 77.4 |
| **Self-supervised** | | | | | | | |
| MAE | ViT-H/14 | INet-1k | ✗ | 49.4 | 76.6 | 83.3 | 64.8 |
| DINO | ViT-S/8 | INet-1k | ✗ | 78.6 | 79.2 | 85.5 | 68.2 |
| SEERv2 | RG10B | IG2B | ✗ | – | 79.8 | – | – |
| MSN | ViT-L/7 | INet-1k | ✗ | 79.2 | 80.7 | 86.0 | 69.7 |
| EsViT | Swin-B/W=14 | INet-1k | ✗ | 79.4 | 81.3 | 87.0 | 70.4 |
| Mugs | ViT-L/16 | INet-1k | ✗ | 80.2 | 82.1 | 86.9 | 70.8 |
| iBOT | ViT-L/16 | INet-22k | ✗ | 72.9 | 82.3 | 87.5 | 72.4 |
| | ViT-S/14 | LVD-142M | ✗ | 79.0 | 81.1 | 86.6 | 70.9 |
| | ViT-B/14 | LVD-142M | ✗ | 82.1 | 84.5 | 88.3 | 75.1 |
| DINOv2 | ViT-L/14 | LVD-142M | ✗ | **83.5** | 86.3 | 89.5 | 78.0 |
| | ViT-g/14 | LVD-142M | ✗ | **83.5** | **86.5** | **89.6** | **78.4** |

Table 4: **Linear evaluation on ImageNet-1k of frozen pretrained features.** We report Top-1 accuracy on the validation set for publicly available models trained on public or private data, and with or without text supervision (text sup.). For reference, we also report the kNN performance on the validation set. We compare across any possible architectures (Arch.), at resolution $224 \times 224$ unless stated otherwise. The dataset used for training EVA-CLIP is a custom mixture, see paper for details (Fang et al., 2023).

arts (91.1%), obtained by Chen et al. (2023a). As DINOv2 leads to features that are strong in both the linear and finetuning settings, a strong property of our approach is that *finetuning is optional*.

| Arch. | Res. | Linear | Finetuned | $\Delta$ |
|-------|------|--------|-----------|----------|
| ViT-g/14 | 224 | 86.5 | 88.5 | +2.0 |
| | 448 | 86.7 | 88.9 | +2.2 |

Table 5: **Supervised finetuning on ImageNet-1k.** We use the pipeline of Touvron et al. (2022) to finetune our encoders on ImageNet-1k at resolutions $224 \times 224$ or $448 \times 448$. We compare with the accuracy obtained with linear probing and observe only modest improvements with fine-tuning: this suggests that DINOv2 features already perform well out-of-the-box.

**Robustness analysis.** To complement our study, and probe the generalization of our features, we evaluate our ImageNet-1k models trained with linear classification heads on domain generalization benchmarks. We use the best performing linear classifier as described above and simply run inference on those benchmarks. Please note that most results in the literature are obtained with models that are finetuned end-to-end on ImageNet-1k. We show the result of this experiment in Table 6. When comparing with state-of-the-art SSL methods, our models shows drastically better robustness (+29.6% on A (Hendrycks et al., 2021b), +22.1% on R (Hendrycks et al., 2021a) and +23.0% on Sketch (Wang et al., 2019) compared to iBOT). Our model also improves upon the best weakly-supervised model on ImageNet-A while lagging behind on R and Sketch.

| Method | Arch | Data | Im-A | Im-R | Im-C↓ | Sketch |
|--------|------|------|------|------|-------|--------|
| OpenCLIP | ViT-G/14 | LAION-2B | 63.8 | **87.8** | 45.3 | **66.4** |
| MAE | ViT-H/14 | INet-1k | 10.2 | 34.4 | 61.4 | 21.9 |
| DINO | ViT-B/8 | INet-1k | 23.9 | 37.0 | 56.6 | 25.5 |
| iBOT | ViT-L/16 | INet-22k | 41.5 | 51.0 | 43.9 | 38.5 |
| | ViT-S/14 | LVD-142M | 33.5 | 53.7 | 54.4 | 41.2 |
| DINOv2 | ViT-B/14 | LVD-142M | 55.1 | 63.3 | 42.7 | 50.6 |
| | ViT-L/14 | LVD-142M | 71.3 | 74.4 | 31.5 | 59.3 |
| | ViT-g/14 | LVD-142M | **75.9** | 78.8 | **28.2** | 62.5 |

Table 6: **Domain Generalization with a linear probe** on top of frozen features at a resolution of 224. Higher numbers are better for all benchmarks except Im-C.

| Feature | Arch | Image classification | | | Video classification | | |
|---------|------|----------|----------|-----------|------|---------|------|
| | | iNat2018 | iNat2021 | Places205 | K400 | UCF-101 | SSv2 |
| OpenCLIP | ViT-G/14 | 73.0 | 76.0 | **69.8** | 78.3 | 90.7 | 35.8 |
| MAE | ViT-H/14 | 31.0 | 32.3 | 52.4 | 54.2 | 70.6 | 29.2 |
| DINO | ViT-B/8 | 59.6 | 68.3 | 60.4 | 64.5 | 85.0 | 32.6 |
| iBOT | ViT-L/16 | 66.3 | 74.6 | 64.4 | 72.6 | 88.6 | **38.7** |
| | ViT-S/14 | 69.0 | 74.2 | 62.9 | 67.8 | 87.0 | 33.1 |
| DINOv2 | ViT-B/14 | 76.4 | 81.1 | 66.2 | 73.2 | 89.1 | 34.4 |
| | ViT-L/14 | 80.4 | 85.1 | 67.3 | 76.3 | 90.5 | 35.6 |
| | ViT-g/14 | **81.6** | **85.7** | 67.5 | **78.4** | **91.2** | 38.3 |

Table 7: **Linear evaluation on other image and video classification.** The image benchmarks contain a large quantity of fine-grained examples about objects or scenes. The video benchmarks cover action classification and human-object interaction. All the features are frozen with a linear probe on top.

## 7.2 Additional Image and Video classification Benchmarks

In this section we study the generalization of our features on downstream classification benchmarks. We consider two sets of evaluations in that context. On one hand, we use large and finegrained datasets such as iNaturalist and Places205. On the other, we use the 12 image classification tasks originally proposed in SimCLR (Chen et al., 2020). For iNaturalist 2018, iNaturalist 2021, and Places205, we train a linear classifier with data augmentations as in Sec. 7.1 We report top-1 accuracy for those three datasets in Table 7. Interestingly, our model significantly outperforms OpenCLIP ViT-G/14 on both variants of iNaturalist (+8.6% and +9.7% for 2018 and 2021 respectively), and lags slightly behind on Places 205 (−2.3%).

In a second set of evaluations, we measure the performance of our model on video action recognition even though our features were not trained on videos.. We evaluated features on three datasets, namely UCF-101 (Soomro et al., 2012), Kinetics-400 (Kay et al., 2017) and Something-Something v2 (Goyal et al., 2017). For this evaluation, we pick 8 evenly spaced frames in the video and train a linear classifier on the average of the features for UCF and K-400. For SSv2, we opt for concatenation to retain more temporal information than with feature averaging. For each dataset, we measure average accuracy and report the results in Table 7. We see that amongst self-supervised approaches, our model clearly sets a new state of the art. Moreover, our model matches the accuracy of the OpenCLIP features on UCF and Kinetics (+0.1% and +0.5% respectively) and clearly outperforms them on SSv2 (+2.5%). This is particularly interesting, as SSv2 requires a much richer understanding of the video frames.

Finally, in Table 8, we compare selected frozen features on 12 transfer classification benchmarks initially proposed by Chen et al. (2020). This benchmark covers scenes, objects (food, cars, planes), and textures.

| Feature | Arch | Food | C10 | C100 | SUN | Cars | Aircr | VOC | DTD | Pets | Cal101 | Flowers | CUB | Avg |
|---------|------|------|-----|------|-----|------|-------|-----|-----|------|--------|---------|-----|-----|
| OpenCLIP | ViT-G/14 | 94.5 | 98.7 | 91.0 | **84.0** | **96.1** | 80.2 | **89.3** | **86.0** | 95.7 | **98.1** | 99.5 | 89.9 | 91.9 |
| MAE | ViT-H/14 | 78.4 | 96.1 | 83.9 | 63.9 | 56.1 | 63.4 | 84.3 | 75.4 | 89.4 | 95.9 | 92.3 | 57.2 | 78.0 |
| DINO | ViT-B/8 | 85.1 | 97.2 | 86.9 | 70.3 | 76.6 | 70.6 | 86.7 | 79.6 | 93.2 | 95.4 | 97.6 | 81.7 | 85.1 |
| iBOT | ViT-L/16 | 91.0 | 99.0 | 92.8 | 75.6 | 71.8 | 72.4 | 89.0 | 80.7 | 87.7 | 97.5 | 99.6 | 82.1 | 86.6 |
| DINOv2 | ViT-S/14 | 89.1 | 97.7 | 87.5 | 74.4 | 81.6 | 74.0 | 87.8 | 80.6 | 95.1 | 97.0 | 99.6 | 88.1 | 87.7 |
| | ViT-B/14 | 92.8 | 98.7 | 91.3 | 77.3 | 88.2 | 79.4 | 88.2 | 83.3 | 96.2 | 96.1 | 99.6 | 89.6 | 90.1 |
| | ViT-L/14 | 94.3 | 99.3 | 93.4 | 78.7 | 90.1 | 81.5 | 88.3 | 84.0 | 96.6 | 97.5 | 99.7 | 90.5 | 91.2 |
| | ViT-g/14 | **94.7** | **99.5** | **94.4** | 78.7 | 91.4 | **87.2** | 89.0 | 84.5 | **96.7** | 97.6 | **99.7** | **91.6** | **92.1** |

Table 8: **Linear evaluation of frozen features on fine-grained benchmarks.** Accuracy on 12 benchmarks covering objects, scenes and textures following the evaluation protocol proposed in Chen et al. (2020).

| | | Oxford | | Paris | | Met | | | AmsterTime |
|---------|------|--------|------|-------|------|-----|------|-----|------------|
| Feature | Arch | M | H | M | H | GAP | GAP- | ACC | mAP |
| OpenCLIP | ViT-G/14 | 50.7 | 19.7 | 79.2 | 60.2 | 6.5 | 23.9 | 34.4 | 24.6 |
| MAE | ViT-H/14 | 11.7 | 2.2 | 19.9 | 4.7 | 7.5 | 23.5 | 30.5 | 4.2 |
| DINO | ViT-B/8 | 40.1 | 13.7 | 65.3 | 35.3 | 17.1 | 37.7 | 43.9 | 24.6 |
| iBOT | ViT-L/16 | 39.0 | 12.7 | 70.7 | 47.0 | 25.1 | 54.8 | 58.2 | 26.7 |
| DINOv2 | ViT-S/14 | 68.8 | 43.2 | 84.6 | 68.5 | 29.4 | 54.3 | 57.7 | 43.5 |
| | ViT-B/14 | 72.9 | 49.5 | 90.3 | 78.6 | 36.7 | 63.5 | 66.1 | 45.6 |
| | ViT-L/14 | **75.1** | **54.0** | **92.7** | **83.5** | **40.0** | 68.9 | 71.6 | **50.0** |
| | ViT-g/14 | 73.6 | 52.3 | 92.1 | 82.6 | 36.8 | **73.6** | **76.5** | 46.7 |

Table 9: **Evaluation of frozen features on instance-level recognition.** We consider 4 different benchmarks and report their main metrics.

We replace the Birdsnap dataset with CUB because the former was not publicly available in its entirety. We follow the experimental protocol as outlined by Chen et al. (2020), namely training a logistic regression on precomputed features. Our model significantly outperforms state-of-the-art SSL models, with most notable differences on Stanford Cars (+14.8% versus DINO ViT-B/8) and FGVC Aircraft (+14.8% versus iBOT ViT-L/16). Even though these benchmarks favor text-guided pretraining, our features are still competitive with OpenCLIP on most classification benchmarks, with the exception of a few datasets, especially SUN (−5.3%) and Cars (−4.7%).

## 7.3 Instance Recognition

In this experiment, we probe our model on the task of instance-level recognition using a non-parametric approach. Images from a database are ranked according to their cosine similarity with a query image. We evaluated our model and compare to baselines on Paris and Oxford, that are landmark recognition benchmarks. We also evaluated on Met, a dataset of artworks from the Metropolitan museum, and AmsterTime, containing street view images matched to archival images of Amsterdam. We measure performance by computing the mean average precision and report our results in Table 9. We see that our features significantly outperform both SSL (+41% mAP on Oxford-Hard), and weakly-supervised (+34% mAP on Oxford-Hard) ones. It is interesting to see that our features perform well across task granularities, both at the category-level and instance-level. This is a desirable property for strong off-the-shelf computer vision features.

| Method | Arch. | ADE20k (62.9) | | CityScapes (86.9) | | Pascal VOC (89.0) | |
|---|---|---|---|---|---|---|---|
| | | lin. | +ms | lin. | +ms | lin. | +ms |
| OpenCLIP | ViT-G/14 | 39.3 | 46.0 | 60.3 | 70.3 | 71.4 | 79.2 |
| MAE | ViT-H/14 | 33.3 | 30.7 | 58.4 | 61.0 | 67.6 | 63.3 |
| DINO | ViT-B/8 | 31.8 | 35.2 | 56.9 | 66.2 | 66.4 | 75.6 |
| iBOT | ViT-L/16 | 44.6 | 47.5 | 64.8 | 74.5 | 82.3 | 84.3 |
| DINOv2 | ViT-S/14 | 44.3 | 47.2 | 66.6 | 77.1 | 81.1 | 82.6 |
| | ViT-B/14 | 47.3 | 51.3 | 69.4 | 80.0 | 82.5 | 84.9 |
| | ViT-L/14 | 47.7 | **53.1** | 70.3 | 80.9 | 82.1 | 86.0 |
| | ViT-g/14 | **49.0** | 53.0 | **71.3** | **81.0** | **83.0** | **86.2** |

Table 10: **Semantic segmentation on ADE20K, CityScapes and Pascal VOC with frozen features** and a linear classifier (lin.) and with multiscale (+ms). The absolute state of the art – from Wang et al. (2022), Liu et al. (2021) and Chen et al. (2018) respectively – are mentioned at the top of the Table. For reference, using the Mask2Former pipeline (Steiner et al., 2021) with a ViT-Adapter (Chen et al., 2023b) on top of our frozen ViT-g/14 backbone gives 60.2 mIoU on ADE-20k.

## 7.4 Dense Recognition Tasks

We probe the quality of patch-level features extracted from our network on several dense downstream tasks. We consider semantic image segmentation and monocular depth estimation in several settings and we conduct evaluations on multiple datasets for each.

**Semantic segmentation.** For our semantic segmentation evaluation, we consider two different setups. **Linear**: a linear layer is trained to predict class logits from a patch tokens. It is used to produce a low-resolution logit map (eg 32x32 for a model with patch size 16), which is then upsampled to full resolution (512x512) to obtain a segmentation map. This procedure is extremely simple but cannot easily produce high-resolution segmentations. **+ms**: a boosted version of the linear setup. We concatenate the patch tokens of the 4 last layers, use a larger image resolution of 640, and use multiscale test-time augmentations to improve the predictions. We report the performance of our model variants as well as the baselines on three datasets under the two setups in Table 10.

Our models show very good performance on all datasets and for all setups. Interestingly, our evaluation using **+ms** is on par with fully finetuning MAE with an Upernet decoder (53.0 versus 53.6 mIoU). This is surprising because we use a significantly simpler predictor. Also, our best model, when evaluated using the boosted recipe, almost matches the state of the art on Pascal VOC (86.2 versus 89.0 mIoU).

**Frozen backbone in a SOTA pipeline.** In a final experiment, we freeze our backbone, and plug it into a ViT-Adapter Chen et al. (2023b) with a Mask2former head (Cheng et al., 2022). We tune the weights of the adapter and head, but keep the backbone frozen, meaning 66% of the weights are frozen. This allows for a lighter segmentation training than full end-to-end fine-tuning. With this setup, we reach 60.2 mIoU on ADE20k, close to the competitive state of the art, standing at 62.9 mIoU (Wang et al., 2022). Although our setup for this experiment doesn't makes use of the optimisations described in Sec. 5, the segmentation training in this experiment took 28 hours on 16 V100 GPUs.

**Depth estimation.** In this experiment, we evaluate our patch-level features on three monocular depth estimation benchmarks: NYUd, KITTI and zero-shot transfer from NYUd to SUN3d. We follow the evaluation protocol of Li et al. (2022b). We consider three different setups for this evaluation. **lin. 1**: we extract the last layer of the frozen transformer and concatenate the [CLS] token to each patch token. Then we bi-linearly upsample the tokens by a factor of 4 to increase the resolution. Finally we train a simple linear layer using a classification loss by dividing the depth prediction range in 256 uniformly distributed bins and

| Method | Arch. | NYUd (0.330) | | | KITTI (2.10) | | | NYUd → SUN RGB-D (0.421) | | |
|---|---|---|---|---|---|---|---|---|---|---|
| | | lin. 1 | lin. 4 | DPT | lin. 1 | lin. 4 | DPT | lin. 1 | lin. 4 | DPT |
| OpenCLIP | ViT-G/14 | 0.541 | 0.510 | 0.414 | 3.57 | 3.21 | 2.56 | 0.537 | 0.476 | 0.408 |
| MAE | ViT-H/14 | 0.517 | 0.483 | 0.415 | 3.66 | 3.26 | 2.59 | 0.545 | 0.523 | 0.506 |
| DINO | ViT-B/8 | 0.555 | 0.539 | 0.492 | 3.81 | 3.56 | 2.74 | 0.553 | 0.541 | 0.520 |
| iBOT | ViT-L/16 | 0.417 | 0.387 | 0.358 | 3.31 | 3.07 | 2.55 | 0.447 | 0.435 | 0.426 |
| DINOv2 | ViT-S/14 | 0.449 | 0.417 | 0.356 | 3.10 | 2.86 | 2.34 | 0.477 | 0.431 | 0.409 |
| | ViT-B/14 | 0.399 | 0.362 | 0.317 | 2.90 | 2.59 | 2.23 | 0.448 | 0.400 | 0.377 |
| | ViT-L/14 | 0.384 | 0.333 | 0.293 | 2.78 | 2.50 | 2.14 | 0.429 | 0.396 | 0.360 |
| | ViT-g/14 | **0.344** | **0.298** | **0.279** | **2.62** | **2.35** | **2.11** | **0.402** | **0.362** | **0.338** |

Table 11: **Depth estimation with frozen features**. We report performance when training a linear classifier on top of one (lin. 1) or four (lin. 4) transformer layers, as well, as the DPT decoder (DPT) of Ranftl et al. (2021). We report the RMSE metric on the 3 datasets. Lower is better. For reference, we report state-of-the-art results taken from Li et al. (2022b) on each benchmark on top of the Table.

use a linear normalization following Bhat et al. (2021). **lin. 4**: we use the same protocol that we use with one layer, but concatenate the tokens from layers $l = \{3, 6, 9, 12\}$ for ViT-S/B, $l = \{5, 12, 18, 24\}$ for ViT-L, and $l = \{10, 20, 30, 40\}$ for ViT-g. **DPT**: we use the DPT decoder (Ranftl et al., 2021) on top of our frozen models and setup a regression task. We scale the size of the head following the dimension of the features for each architecture. We show results for all baselines, all datasets and all setups in Table 11.

From this table, we see that our features clearly surpass the best SSL and WSL features available. It is interesting to see that iBOT features extracted from a ViT-L outperform the ones of OpenCLIP with a ViT-G. This observation supports the intuition that caption-based feature learning fails to learn subtle patterns like this one. Also, our model, with the DPT decoder and frozen backbone, matches or exceeds the performance of the recent work of Li et al. (2022b). Finally, the out-of-domain generalization result on SUN-RGBd shows that our features allow very good transfer between domains. A depth prediction module trained on indoor scenes from NYUd generalizes pretty well to the outdoor examples of SUN-RGBd.

## 7.5 Qualitative Results

In this final section of the empirical evaluation of our features, we propose a few qualitative analyses.

**Semantic Segmentation and Depth Estimation.** We show some qualitative results for our dense prediction evaluations: segmentation on ADE20K in Fig. 7 and depth estimation on NYUd, KITTI and SUN RGB-D in Fig. 7. We compare DINOv2 with OpenCLIP with a linear classifier on each dataset. While not perfect, the linear segmentation model using our DINOv2 backbone produces good results and behaves much better than the OpenCLIP one under this evaluation setup. Indeed, the segmentation mask produced by OpenCLIP-G shows many artifacts and disconnected components. The qualitative results on depth estimation clearly illustrate the quantitative gap between OpenCLIP and DINOv2. These results highlight that our features, as well as the features extracted from OpenCLIP, are able to linearly separate complex information such as depth, even though neither was trained with this type of information. However, our features lead to a much smoother depth estimation, with less artifacts. Some objects such as the chair on the SUN RGB-D image are completely ignored by OpenCLIP and correctly positioned using our features.

**Out-of-distribution generalization.** We show a few examples of applying the depth prediction and segmentation linear classifiers to out-of-distribution examples in Fig. 8. The qualitative results support our claim that our features transfer between domains. The quality of the depth and segmentation predicted for pictures of animals, or paintings is very good, even though the domains are very different.

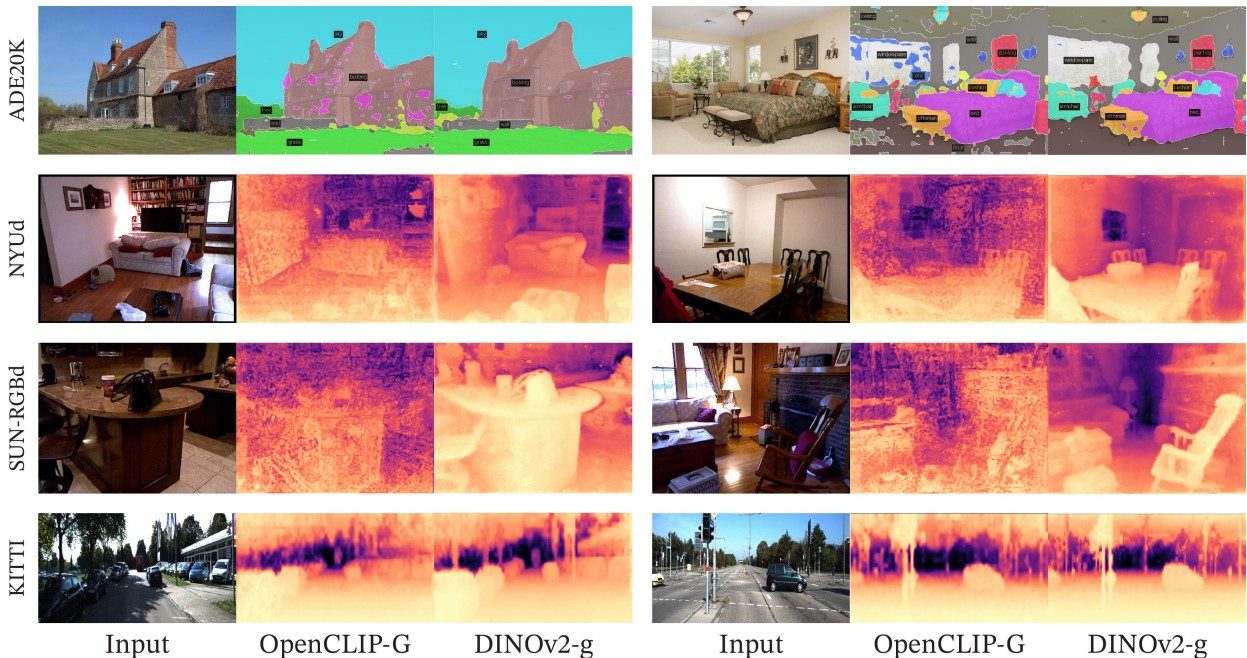

Figure 7: **Segmentation and depth estimation with linear classifiers.** Examples from ADE20K, NYUd, SUN RGB-D and KITTI with a linear probe on frozen OpenCLIP-G and DINOv2-g features.

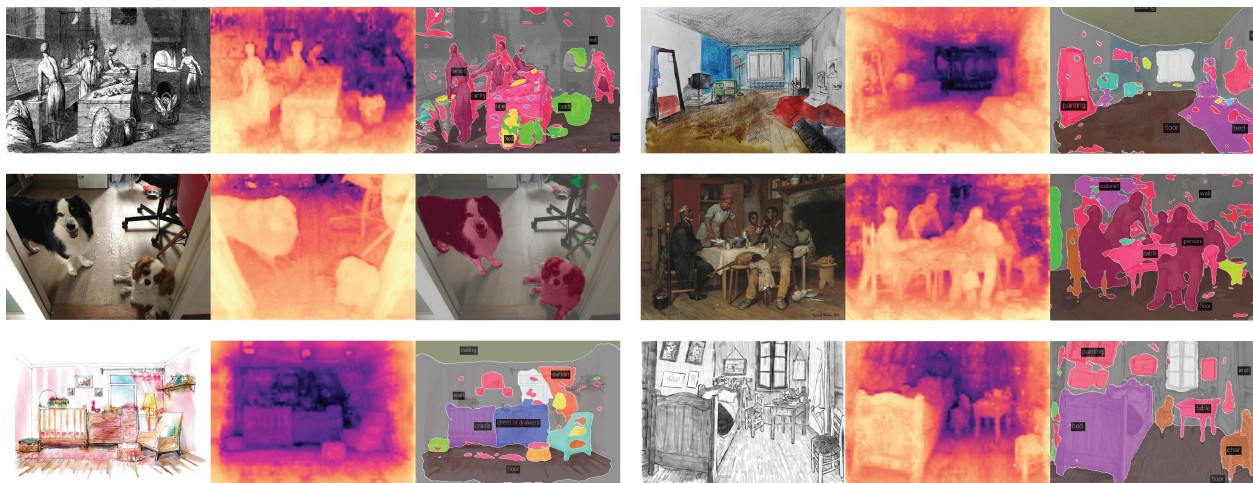

Figure 8: **Examples of out-of-distribution examples** with frozen DINOv2-g features and a linear probe.

**PCA of patch features.** We show the results of the principal component analysis (PCA) performed on the patch features extracted by our model. We keep only patches with a positive value after we threshold the first component. This procedure turns out to separate the image's main object from the background. We compute a second PCA on the remaining patches across three images depicting the same category. We color the three first components with three different colors and present the results in Fig. 1 and 9. There are two interesting observations: first, our unsupervised foreground / background detector, based on detecting the highest variance direction, performs very well and is capable of delineating the boundary of the main object in the picture. Second, the other components correspond to "parts" of objects and match well for images of the same category. This is an emerging property – our model was not trained to parse parts of objects.

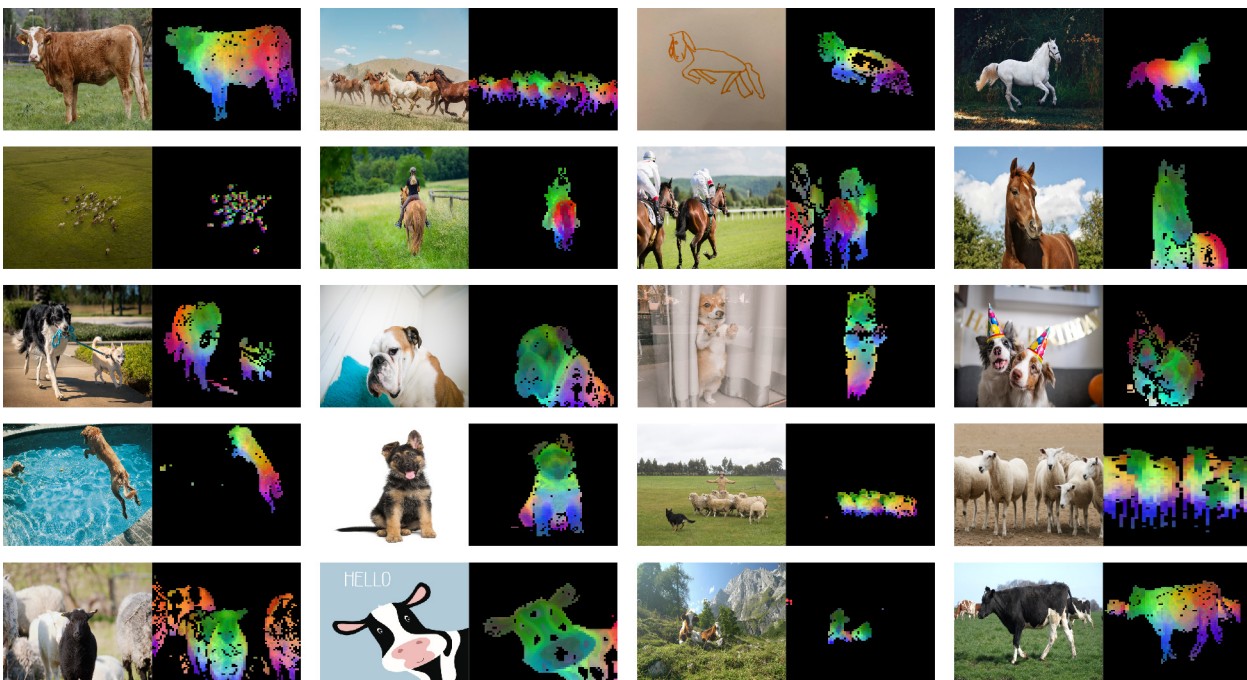

Figure 9: **More visualization of the first PCA components.** We compute the PCA between the patches from all of the images and show their first 3 components. Each component corresponds to a specific color channel. Same parts are matched between related images depsite changes of pose, style or even objects. Background is removed by removing patches with a negative score of the first PCA component.

**Patch matching.** Finally, we explore what type of information our patch-level features contain by matching them across images. We start by detecting the foreground object using the procedure described above. Then, we compute the euclidean distance between patch features extracted from two images and map them by solving an assignment problem. In order to reduce the number of matches, we then apply a non-maximum suppression to keep only the salient ones. In Fig. 10, we show some examples of such matchings.

We observe that the features seem to capture information about semantic regions that serve similar purpose in different objects or animals. For instance, the wing of a plane matches the wing of a bird. We also observe that the model is robust to style (image versus drawing), and to large variation of poses (see the elephant).

# 8 Fairness and Bias Analysis

We conduct two fairness evaluations of our models. We probe for geographical fairness and potential harmful label associations. For both evaluations, we experiment with our largest ViT-g model.

## 8.1 Geographical Fairness

We evaluate geographical fairness on the Dollar Street dataset introduced in De Vries et al. (2019) using the evaluation protocol of Goyal et al. (2022b). This benchmark compares performance across countries and income levels. It contains 16,073 images from 289 households across 54 countries. The task is to recognize 94 concepts that vary visually between households based on income or location. In Table 12, we compare our model with SEERv2 (Goyal et al., 2022a), a model trained on a geographically diverse set of images. Our model is slightly fairer across regions and incomes than the SEERv2 model and significantly better than the supervised baseline reported by Goyal et al. (2022a). However, we still observe a significant difference between regions, particularly in Africa, where our model performance drops by 25.7% compared to Europe. This shows that our model is still biased toward Western countries. Similarly, our model performs

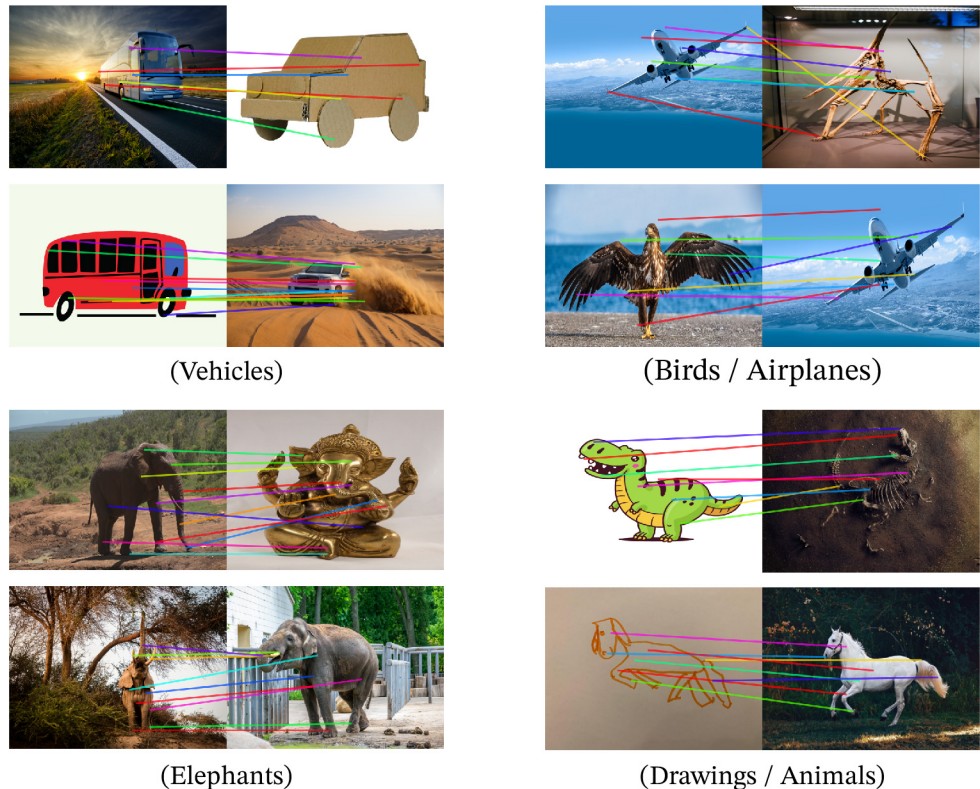

Figure 10: **Matching across images.** We match patch-level features between images from different domains, poses and even objects that share similar semantic information. This exhibits the ability of our model to transfer across domains and understand relations between similar parts of different objects.

| Method | Arch. | Data | Income buckets | | | Regions | | | |
|--------|-------|------|-----|--------|------|--------|------|----------|--------|
| | | | low | medium | high | Africa | Asia | Americas | Europe |
| SEERv2 | RG-10B | IG-1B | 59.7 | 78.5 | 86.6 | 65.9 | 76.3 | 81.1 | 85.6 |
| DINOv2 | ViT-g/14 | LVD-142M | 67.4 | 83.3 | 90.5 | 74.0 | 81.6 | 86.2 | 89.7 |

Table 12: **Geographical fairness and diversity analysis across income buckets and regions.**

significantly better on high-income households than low-income ones, with a difference of 31.7%. Despite improvements, we observe significant biases in our models toward wealthy households from Western countries.

## 8.2 Gender, Skintones and Age

In a second set of evaluations, we question how our model classifies images of people of different gender, skin tone, and age (all self-reported). We follow the protocol of Goyal et al. (2022b), where we train a multiclass classifier on a subset of 619 classes of ImageNet-22k. We group the 619 classes into four broader categories: Human, Possibly Human, Non-Human, or Crime. Non-Human and Crime are considered harmful. Using this classifier, we run inference on 2955 images from the Casual Conversations dataset (Hazirbas et al., 2021) and keep all labels in the top-5 that are assigned a probability of 0.1 or more. Because of that, we can assign multiple classes to each image. We make one modification to the original evaluation protocol: we do not backpropagate gradients to the backbone and keep it frozen. We compare our model to SEERv2 in Table 13.

| Model | Assoc. | Gender Skintone | | | | Age Groups | | | |
|---|---|---|---|---|---|---|---|---|---|
| | | female darker | female lighter | male darker | male lighter | 18-30 | 30-45 | 45-70 | 70+ |
| SEER | Non-Human | 0.0 | 0.0 | 0.0 | 0.0 | 0.0 | 0.0 | 0.0 | 0.0 |
| RG-10B | Crime | 0.0 | 0.0 | 0.0 | 0.0 | 0.0 | 0.0 | 0.0 | 0.0 |
| | Human | 94.9 | 95.8 | 86.6 | 79.0 | 90.5 | 88.3 | 91.9 | 82.3 |
| | Possibly-Human | 13.6 | 6.7 | 65.0 | 60.2 | 32.8 | 37.2 | 29.4 | 6.5 |
| DINOv2 | Non-Human | 0.0 | 0.0 | 0.0 | 0.0 | 0.0 | 0.0 | 0.0 | 0.0 |
| ViT-g/14 | Crime | 0.0 | 0.0 | 0.2 | 0.0 | 0.0 | 0.1 | 0.0 | 0.0 |
| | Human | 97.3 | 97.7 | 86.1 | 84.0 | 91.2 | 90.2 | 93.2 | 88.7 |
| | Possibly-Human | 15.8 | 17.2 | 52.2 | 48.1 | 35.3 | 37.3 | 23.0 | 9.7 |

Table 13: **Label association fairness evaluation across gender, skintones and age groups.** We follow the protocol proposed by Goyal et al. (2022b) with a slight modification. Instead of finetuning the backbone, we simply learn a linear classifier on the subset of 619 classes of ImageNet-22k.

| Model to Reproduce | GPU Type | GPU Power consumption | GPU-hours | PUE | Total power consumption | Carbon emitted (tCO$_2$eq) |
|---|---|---|---|---|---|---|
| DINOv2-g | A100-40GB | 400W | 22,016 | 1.1 | 9.7 MWh | 3.7 |

Table 14: **Carbon footprint of reproducing DINOv2.** We report the potential carbon emission of reproducing DINOv2-g when assuming a power consumption for the A100-40GB of 400W, a PUE of 1.1 and carbon intensity factor of 0.385 kg CO$_2$e per KWh.

Our model often classifies images of all groups as Human without large deviations across skin tones. Neither SEERv2 nor DINOv2 predict harmful labels from the Non-Human or Crime meta-categories (except for two instances where the background contains bars visually similar to prison bars). We see that our model triggers the Possibly-Human classes often. This class is constructed from objects in ImageNet-22k that are often related to Humans, such as Scarf, Glasses, or Beard. Our model often predicts the Possibly-Human class for men because of the prevalence of the Beard class. No clear pattern indicates a bias against a particular group in this study. While this is encouraging, we also acknowledge that a more thorough evaluation of biases may reveal flaws in our model.

## 9 Estimating the Environmental Impact of Training our Models

Training foundation models consumes a significant amount of energy, resulting in carbon dioxide emissions. Patterson et al. (2021) propose a methodology to report an estimation of the carbon emitted during the training of a model based on the specifics of the data center and its power grid. This computation informs the design of the data center used for the training of models and the choice of location for data centers. This methodology requires to know the specifics of the data center used for training, which can be complex when multiple data centers are involved over time. Additionally, these specifics are most often not in the control of the AI practitioner, and hence, this methodology is less helpful when practioners make technical decisions about future trainings. Instead, in this section, we follow an alternative that reports the potential carbon emission of retraining a similar model in an average data center located in the US. This methodology was used in previous work in natural language processing (Strubell et al., 2019; Touvron et al., 2023) to establish an apple-to-apple comparison between pretraining schemes. More precisely, we fix the value of all exogenous variables, i.e., the Power Usage Effectiveness (PUE) and carbon intensity factor of a power grid to the same values as in Touvron et al. (2023), that is, a PUE of 1.1 and the carbon intensity factor to the US average of 0.385 kg CO$_2$eq/KWh. We use the same formula as in Patterson et al. (2021) to estimate the potential energy consumption and the carbon emission. For the power consumption of an A100-80GB, we take the thermal design power for NVLink systems, which is 400W. We report the potential carbon emission

of retraining a DINOv2 ViT-g in Table 14. For comparison, retraining an OpenCLIP ViT-L or OpenCLIP ViT-G would require 22.4 MWh and 118.9 MWh, respectively, if run in the same data center. This is $10\times$ more carbon emission. Note that this comparison is not fair to them, since they also train a text encoder in parallel, and we thus do not report them in the table. However, it gives a reasonable guideline for those who are interested in training only visual features: in this context, training a self-supervised model is preferable in terms of carbon emission. Training a text-guided model still makes sense when planning to reuse the text encoder.

**Carbon footprint of the whole project.**    Additionally, we estimate the footprint of the whole project to be between 0.5k and 1k tCO$_2$eq using the same grid as presented above [3]. This carbon footprint represents in the order of 200k GPU-days. The primary sources of emissions are the self-supervised pre-trainings of the models. For example, a single pre-training of a ViT-g model (22k GPU-hours) emits 3.7 tons of CO$_2$eq, while a finetuning on ImageNet-1k (1k GPU-hours) emits 0.2 tons. This estimate only considers the GPUs' electricity consumption and ignores other emissions, such as their manufacturing and disposal.

## 10   Future work and Discussion

In this work, we present DINOv2, a new series of image encoders pretrained on large curated data with no supervision. This is the first SSL work on image data that leads to visual features that close the performance gap with (weakly) supervised alternatives across a wide range of benchmarks and without the need for finetuning.   We can attribute the strong performance of the DINOv2 family of models to several factors: **i**) an improved training recipe with better hyperparameters and regularization (Table 1), **ii**) a larger model scale with improved results regardless of the data used for training (Fig. 4), **iii**) a larger dataset (Fig. 4) and **iv**) the distillation process that makes smaller models benefit from the performance of the strongest ViT-g model (Fig. 5).   A few properties emerge from these models, such as an understanding of object parts and scene geometry regardless of the image domains. We expect that more of these properties will emerge at larger scales of models and data, akin to instruction emergence in large language models, and plan to continue scaling along these axes. This paper also demonstrates that these visual features are compatible with classifiers as simple as linear layers - meaning the underlying information is *readily available.* In future work, we plan to leverage this ability to train a a language-enabled AI system that can process visual features as if they were word tokens, and extract the required information to ground the system.

**Acknowledgments.**

We thank Mathilde Caron for initial discussions that led to this work. Julien Mairal was supported by the ERC grant number 101087696 (APHELAIA project) and by ANR 3IA MIAI@Grenoble Alpes (ANR-19-P3IA-0003). We thank Olivia Joulin for the horse drawing used in Fig. 10. We thank Madeleine and Léon for posing for Fig. 8 We also thank the rest of FAIR and Meta AI for feedback on this work through the entire project.

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

# A Data Processing

## A.1 Data selection

Our selection of datasets for building LVD-142M is detailed in Tab. 15. This collection is intended to provide images covering well various downstream vision tasks both for image-level and dense recognition.

## A.2 Image similarity

We employ cosine similarity to compare image features (whether ours or feature generated for deduplication) with the following similarity function $m$:

$$m(s,r) = \text{cosine-similarity}\left(f\left(s\right), f\left(r\right)\right) = \frac{f(s) \cdot f(r)}{\|f(s)\|_2 \|f(r)\|_2}$$

where $s$ and $r$ are a pair of images to compare and $f$ is the model generating features.

## A.3 Deduplication

**Self-deduplication.** To deduplicate our uncurated data source of 1.3B images, we compute and use the embeddings generated by Pizzi et al. (2022) and retrieve the $k = 64$ nearest neighbors of each image (using cosine similarity). Considering only neighbors with a similarity $>0.6$, we extract the connected components of the associated $k$-NN graph thanks to a scalable disjoint set data structure implementation. We then only keep one representative for each component of duplicate images. This results in a self-deduplicated data source of 1.1B images.

**Relative deduplication** To reduce redundancy and also properly evaluate the performance of our features, we discard remaining images of our self-deduplicated data source that are too similar to train and test splits of our evaluation datasets. To achieve this, we apply a similar procedure as for self-deduplication, with a stricter similarity $>0.45$, this time identifying the duplicate components (if any) to which each reference image belong and discarding it entirely. This results in a self- and relatively-deduplicated data source of 744M images.

## A.4 Retrieval

We employ two approaches to augment dataset via retrieval: sample-based and cluster-based. The first one, sample-based, applies to datasets larger than 1M images and consists in collecting a fixed number $k$ of nearest images for each sample image of the dataset to retrieve, effectively trying to multiply by $k$ the size of the dataset. We use $k = 4$ for Google Landmarks v2 and ImageNet-22k but a larger $k = 32$ to make this specific retrieval a core part of our LVD-142M dataset. For smaller datasets, the second approach, cluster-based, consists in first clustering our uncurated data source into $100,000$ separate clusters thanks to a distributed $k$-means implementation. Each cluster should capture different types of image concept and contents. We then pick $10,000$ images from each cluster associated with more than 3 images of the retrieved dataset. As this can result in a very large number of retrieved images for some dataset, we restrict such retrievals to a maximum of 1M images to maintain the balance between the different datasets within LVD-142M.

# B Implementation Details

## B.1 Unsupervised pre-training

For unsupervised pre-training we build on the DINO and iBOT codebases. We use hyperparameters shown in Table 16, ViT architectures described in Table 17.

**KoLeo regularization.** We apply the KoLeo regularizer with a weight of 0.1 between the class tokens of the first global crop, for all samples within a GPU without cross-communication for this step.

| Task | Dataset / Split | Images | Retrieval | Retrieved | Final |
|------|-----------------|-------:|-----------|----------:|------:|
| classification | ImageNet-22k / – | 14,197,086 | as is | – | 14,197,086 |
| classification | ImageNet-22k / – | 14,197,086 | sample | 56,788,344 | 56,788,344 |
| classification | ImageNet-1k / train | 1,281,167 | sample | 40,997,344 | 40,997,344 |
| fine-grained classif. | Caltech 101 / train | 3,030 | cluster | 2,630,000 | 1,000,000 |
| fine-grained classif. | CUB-200-2011 / train | 5,994 | cluster | 1,300,000 | 1,000,000 |
| fine-grained classif. | DTD / train1 | 1,880 | cluster | 1,580,000 | 1,000,000 |
| fine-grained classif. | FGVC-Aircraft / train | 3,334 | cluster | 1,170,000 | 1,000,000 |
| fine-grained classif. | Flowers-102 / train | 1,020 | cluster | 1,060,000 | 1,000,000 |
| fine-grained classif. | Food-101 / train | 75,750 | cluster | 21,670,000 | 1,000,000 |
| fine-grained classif. | Oxford-IIIT Pet / trainval | 3,680 | cluster | 2,750,000 | 1,000,000 |
| fine-grained classif. | Stanford Cars / train | 8,144 | cluster | 7,220,000 | 1,000,000 |
| fine-grained classif. | SUN397 / train1 | 19,850 | cluster | 18,950,000 | 1,000,000 |
| fine-grained classif. | Pascal VOC 2007 / train | 2,501 | cluster | 1,010,000 | 1,000,000 |
| segmentation | ADE20K / train | 20,210 | cluster | 20,720,000 | 1,000,000 |
| segmentation | Cityscapes / train | 2,975 | cluster | 1,390,000 | 1,000,000 |
| segmentation | Pascal VOC 2012 (seg.) / trainaug | 1,464 | cluster | 10,140,000 | 1,000,000 |
| depth estimation | Mapillary SLS / train | 1,434,262 | as is | – | 1,434,262 |
| depth estimation | KITTI / train (Eigen) | 23,158 | cluster | 3,700,000 | 1,000,000 |
| depth estimation | NYU Depth V2 / train | 24,231 | cluster | 10,850,000 | 1,000,000 |
| depth estimation | SUN RGB-D / train | 4,829 | cluster | 4,870,000 | 1,000,000 |
| retrieval | Google Landmarks v2 / train (clean) | 1,580,470 | as is | – | 1,580,470 |
| retrieval | Google Landmarks v2 / train (clean) | 1,580,470 | sample | 6,321,880 | 6,321,880 |
| retrieval | AmsterTime / new | 1,231 | cluster | 960,000 | 960,000 |
| retrieval | AmsterTime / old | 1,231 | cluster | 830,000 | 830,000 |
| retrieval | Met / train | 397,121 | cluster | 62,860,000 | 1,000,000 |
| retrieval | Revisiting Oxford / base | 4,993 | cluster | 3,680,000 | 1,000,000 |
| retrieval | Revisiting Paris / base | 6,322 | cluster | 3,660,000 | 1,000,000 |
| | | | | | 142,109,386 |

Table 15: **Composition of our LVD-142M dataset.** We report the list of datasets and associated splits used to build the dataset, how they were included (as is without retrieval or via sample-based or cluster-based retrieval). For retrievals, we indicate the actual number of retrieved images and the final number included in the dataset. We chose to include as many datasets as possible in the pretraining data in order to cover as many domains as possible. We kept a few datasets aside in order to evaluate performance outside of the pretraining domain. More details about dataset usages can be found in Table 18.

|  | Arch. | Drop-rate | LR | Batch size |
|---|---|---|---|---|
| DINOv2-S (distilled) | ViT-S/14 | 0 | 1e-3 | 2048 |
| DINOv2-B (distilled) | ViT-B/14 | 0 | 1e-3 | 2048 |
| DINOv2-L (distilled) | ViT-L/14 | 0 | 1e-3 | 2048 |
| DINOv2-L (from scratch) | ViT-L/14 | 0.4 | 3.5e-4 | 3072 |
| DINOv2-g (from scratch) | ViT-g/14 | 0.4 | 3.5e-4 | 3072 |

Table 16: **Training hyperparameters for DINOv2-S, DINOv2-B, DINOv2-L and DINOv2-g.** All models run for 625k iterations with optimizer AdamW, an initial LayerScale value of 1e-5, a weight decay cosine schedule from 0.04 to 0.2, a learning rate warmup of 100k iterations, a teacher momentum cosine schedule from 0.994 to 1, and we train in float16 precision in all cases (except for the DINO heads where we reduce the gradients in float32).

| Arch. | Embed dim | Heads | Blocks | FFN layer |
|---|---|---|---|---|
| ViT-S/14 (distilled) | 384 | 6 | 12 | MLP |
| ViT-B/14 (distilled) | 768 | 12 | 18 | MLP |
| ViT-L/14 (distilled) | 1024 | 16 | 24 | MLP |
| ViT-L/14 (from scratch) | 1024 | 16 | 24 | SwiGLU |
| ViT-g/14 (from scratch) | 1536 | 24 | 40 | SwiGLU |

Table 17: **Architecture details of the ViT-S/B/L/g networks used in this work.** We use MLP feed-forward networks for distilled models, and SwiGLU (Shazeer, 2020) when training from scratch.

**EMA update for the teacher.** The teacher is initialized with the same state as the student, and is an exponential moving average of the student network, with a momentum value in [0.994, 1.0] following a cosine schedule. It is updated at the end of every training step.

### B.2 High-Resolution adaptation

We initialise the model with the pretrained weights then train it for 10k iterations with the same procedure as the original pretraining. All the schedules are kept the same as in the original training, but compressed to fit in 10k iterations. All the hyperparameters are kept the same as in the first pretraining, except the base learning rate which is reduced.

### B.3 Linear probing evaluation

For linear probing we define 3 evaluation parameters: the learning rate, how many output layers we use, whether we concatenate the average-pooled patch token features with the class token (or use only the class token). We train our linear layer with SGD for 12500 iterations, using random-resized-crop data augmentation, and perform the following grid search:

- learning rate in $\{0.0001, 0.0002, 0.0005, 0.001, 0.002, 0.005, 0.01, 0.02, 0.05, 0.1, 0.2, 0.3, 0.5\}$
- output layers in $\{1, 4\}$
- concatenate average-pooled tokens in $\{yes, no\}$

We then report the highest accuracy value obtained on the validation set as is common practice. Note that this grid search is not expensive, because at each iteration we perform inference on the backbone only once, then feed the output to all linear classifiers (each performing a single matrix multiplication).

## C List of Datasets used

We show in Table 18 the list of benchmarks and datasets used and their purposes.

| Dataset | Pretraining (as is) | Retrieving pretraining data | Eval. | Task | Citation |
|---|---|---|---|---|---|
| ImageNet-1k | ✗ | ✓ | ✓ | Classif. | (Russakovsky et al., 2015) |
| ImageNet-22k | ✓ | ✓ | ✗ | | (Deng et al., 2009) |
| ImageNet-V2 | ✗ | ✗ | ✓ | Classif. | (Recht et al., 2019) |
| ImageNet-ReaL | ✗ | ✗ | ✓ | Classif. | (Beyer et al., 2020) |
| ImageNet-A | ✗ | ✗ | ✓ | Classif. | (Hendrycks et al., 2021b) |
| ImageNet-C | ✗ | ✗ | ✓ | Classif. | (Hendrycks & Dietterich, 2019) |
| ImageNet-R | ✗ | ✗ | ✓ | Classif. | (Hendrycks et al., 2021a) |
| ImageNet-Sk. | ✗ | ✗ | ✓ | Classif. | (Wang et al., 2019) |
| Food-101 | ✗ | ✓ | ✓ | Classif. | (Bossard et al., 2014) |
| CIFAR-10 | ✗ | ✓ | ✓ | Classif. | (Krizhevsky et al., 2009) |
| CIFAR-100 | ✗ | ✓ | ✓ | Classif. | (Krizhevsky et al., 2009) |
| SUN397 | ✗ | ✓ | ✓ | Classif. | (Xiao et al., 2010) |
| StanfordCars | ✗ | ✓ | ✓ | Classif. | (Krause et al., 2013) |
| FGVC-Aircraft | ✗ | ✓ | ✓ | Classif. | (Maji et al., 2013) |
| VOC 2007 | ✗ | ✓ | ✓ | Classif. | (Everingham et al., 2010) |
| DTD | ✗ | ✓ | ✓ | Classif. | (Cimpoi et al., 2014) |
| Oxford Pets | ✗ | ✓ | ✓ | Classif. | (Parkhi et al., 2012) |
| Caltech101 | ✗ | ✓ | ✓ | Classif. | (Fei-Fei et al., 2004) |
| Flowers | ✗ | ✓ | ✓ | Classif. | (Nilsback & Zisserman, 2008) |
| CUB200 | ✗ | ✓ | ✓ | Classif. | (Welinder et al., 2010) |
| iNaturalist 2018 | ✗ | ✗ | ✓ | Classif. | (Van Horn et al., 2018) |
| iNaturalist 2021 | ✗ | ✗ | ✓ | Classif. | (Van Horn et al., 2021) |
| Places-205 | ✗ | ✗ | ✓ | Classif. | (Zhou et al., 2014) |
| UCF101 | ✗ | ✗ | ✓ | Video | (Soomro et al., 2012) |
| Kinetics-400 | ✗ | ✗ | ✓ | Video | (Kay et al., 2017) |
| SSv2 | ✗ | ✗ | ✓ | Video | (Goyal et al., 2017) |
| GLD v2 | ✓ | ✓ | ✗ | | (Weyand et al., 2020) |
| R-Paris | ✗ | ✓ | ✓ | Retrieval | (Radenović et al., 2018a) |
| R-Oxford | ✗ | ✓ | ✓ | Retrieval | (Radenović et al., 2018a) |
| Met | ✗ | ✓ | ✓ | Retrieval | (Ypsilantis et al., 2021) |
| Amstertime | ✗ | ✓ | ✓ | Retrieval | (Yildiz et al., 2022) |
| ADE20k | ✗ | ✓ | ✓ | Seg. | (Zhou et al., 2017) |
| Cityscapes | ✗ | ✓ | ✓ | Seg. | (Cordts et al., 2016) |
| VOC 2012 | ✗ | ✓ | ✓ | Seg. | (Everingham et al., 2010) |
| Mapillary SLS | ✓ | ✗ | ✗ | | (Warburg et al., 2020) |
| NYU-Depth V2 | ✗ | ✓ | ✓ | Depth | (Silberman et al., 2012) |
| KITTI | ✗ | ✓ | ✓ | Depth | (Geiger et al., 2013) |
| SUN-RGBD | ✗ | ✓ | ✓ | Depth | (Song et al., 2015) |
| DollarStreet | ✗ | ✗ | ✓ | Fairness | (De Vries et al., 2019) |
| Casual Conv. | ✗ | ✗ | ✓ | Fairness | (Hazirbas et al., 2021) |

Table 18: **List of datasets used.**

