# OpenReview forum: "DINOv2: Learning Robust Visual Features without Supervision"
_TMLR — Accepted by TMLR_

### Review · Reviewer_SnYd · 2023-08-10

**Summary Of Contributions:**

The work is a follow-up and an expansion of the Dino method with several improvements. It scales Dino to a larger dataset, and accelerates and stabilizes training at scale. The authors also propose a new pretraining dataset (LVD-142M), which is diverse and curated in contrast to the uncurated datasets commonly used in SSL training. The authors benchmark their models on a wide range of datasets and report SotA results on many of them. Notably, their method compares favorably to models using images and captions, e.g. CLIP, while DinoV2 only uses images.

**Audience:**

Yes

**Broader Impact Concerns:**

I think the training data (the curated LVD-142M dataset) should be released such that dataset biases can be assessed by the community. The foundation models released by the authors will be used by many researchers for many different tasks, and as a community, we should have the ability to study all possible sources of biases and un-fairness inherited by these models.

**Claims And Evidence:**

Yes

**Requested Changes:**

I have written down several questions to the authors in the Requested Changes section. Many points are rather questions than requested changes, and I tried to mark them accordingly.

- RQ: requested change
- Q: question

[RQ, please provide intuition for the results if possible] In general, I think it is not very clear what it is about DinoV2 that leads to its strong performance. Why does it perform so well?
- “A few properties emerge from these models, such as an understanding of object parts and scene geometry regardless of the image domains.” -> Do you have an understanding which part of the loss / the architecture/ the training paradigm is responsible for this emergent behavior?
- Looking at Tables 6 and 7, do the authors think the superior performance stems from the better dataset (Inet-22k vs LVD-142M) or from using a better objective compared to e.g. iBOT?

[major, RQ]: section 6.2, page 7: “When compared with models trained on ImageNet-22k, training on LVD-142M is also superior on all the benchmarks but ImageNet-1k. This confirms that training on a more diverse set of images improves the quality of the features in **domains that are not covered by this dataset.**” -> The latter statement is very questionable since both ImageNet1K and ADE-20k have been used to curate LVD-142M. LVD-142M has been designed to contain images that are most similar to these datasets, thus, the claim that these domains are not covered by LVD-142M is wrong, in my view. Please reformulate this sentence or use a holdout set of test datasets that were not used for curation. Also, what is Oxford-M? I couldn’t find it in the Tables 15/18.

[Q]: Following up on the last point, we know that IN-22k was also used to filter the uncurated dataset to build LVD-142M. The performance when training on LVD-142M is fairly similar to when training on IN-22k. I wonder whether the images that are most similar to IN-22k are causal for the increase in robustness compared to the uncurated dataset.

### Data preprocessing:

[Q] “For the uncurated data source, we collect a raw unfiltered dataset of images from a publicly available repository of crawled web data. From each web page in the repository, we extract URL links of images from <img> tags. We discards URLs that are unsafe or restricted by domains, and post-process the downloaded images (PCA hash deduplication, NSFW filtering, and blurring identifiable faces). This results in 1.2B unique images.” -> 1) Will these URL be made publicly available? 2) If yes, will the code to download them will be made publicly available? 3) Will the deduplication code for the dataset obtained from the image URLs be made publicly available? 4) Do the authors have an idea how much the uncurated dataset intersects with LAION2B? Why didn’t the authors use LAION2B as their starting point?

[Q/ RQ] “We build our curated pretraining dataset by retrieving images from our uncurated data source that are close to images in our curated sources.” Several questions here:

1)	[**important**, RQ] Does it matter for the evaluation performance whether a dataset has or has not been used for curation? In the results section, it is very hard to see whether a dataset has or has not been used for curation. For example, in Table 2, as far as I understand, all datasets have been used for curation except for ImageNet-A. Same for Figure 4 and other Tables/ Figures. Intuitively, the performance should improve much more on datasets which were used for curation compared to those that were not. Please investigate whether this seems to be the case or not. Maybe reporting averaged accuracies on datasets that were vs were not used for curation on the models shown in Table 2 would be sufficient. If indeed, the performance increase on the datasets used for curation is stronger than on datasets not used for curation, it should be clearly visible in all Tables / Figures which datasets have and have not been used for curation. And then it should also be discussed as a limitation of this method.

2) [RQ, not relevant for acceptance] Would it work to test DinoV2 on the WILDS benchmark? This should be OOD compared to the image datasets used for curation. In general, I don’t expect this to be a huge issue, the results in Table 7 on i-Naturalist and Places-205 which were not used for curation, are strong. This point is not crucial for acceptance because the authors already present a lot of results on different benchmarks.

3) [**important**, RQ, please provide details in the paper] How did you choose which datasets should be part of the curation pipeline and which should only be evaluated on? Was this an iterative process where more and more datasets were added or was the list final before any models were trained?

[RQ, please provide details in the paper] “Given a query dataset for retrieval, if it is large enough we retrieve N (typically 4) nearest
neighbors for each query image.” -> Why 4? Has this choice been validated or is it an arbitrary number?

[RQ, not important for acceptance] Please evaluate on ObjectNet. This dataset should not overlap with neither IN-22k nor with LVD-142M and might provide another interesting point estimate of generalization performance.


There are some typos in the text which I outlined below. I have likely missed a few. A round of proof-reading could help eliminate all typos.


#### Typos [RQ]:
-	Introduction: „additionnally“
-	Introduction: „all-purposed features“ sounds weird to me. I think “general purpose features” sounds better and would suggest to replace all instances of „all-purposed features“ with my suggestion.
-	Page 4: “We discards URLs” -> discard
-	Page 7: “We report the Top-1 accuracy on the validation set of ImageNet-1k with a k-NN
-	and a linear linear in Table 1.”
-	Page 9: “approximate” -> approximately
-	Page 17: “This show that” -> shows
-	Page 19: “it gives a reasonable guidelines” -> guideline



**Strengths And Weaknesses:**

The paper is well written and easy to follow. Clearly, a lot of work and effort went into writing this paper as well as into generating and presenting the results. All in all, this is a very solid submission.
### Strengths:
**Introduction and Related Work**: The motivation of this paper is clearly written in the introduction. The related work section is well structured and comprehensive. I appreciate the authors putting their paper into context at the end of each paragraph in the related work section.

**Methodology**: The authors provide a very detailed description of their method and the loss functions they included in Section 4.  In Table 1, they ablate the different contributions separately, showing that they all improve the final result. The authors also describe the data processing pipeline in a lot of detail, as well as the steps they took to make the approach more efficient.

**Results**: This paper shows a ton of results on various benchmarks, both qualitative and quantitative. The Figures and Tables are all very well designed. Clearly, a lot of time went into how the results should be displayed and presented, and the general quality of the presentation is commendable. For example, the design of Fig. 5 is pretty cool. The structure of the results Section(s) is comprehensive, which is also commendable, given how many results there are.

**Fairness and Bias Analysis**: Given that the authors propose a new foundation model, I appreciate their effort on evaluating its fairness. The choice of benchmark as well as the baseline model they compare to looks reasonable to me.

**Estimating the Environmental Impact of Training our Models**: Similar to the point above, I appreciate the detailed analysis of the environmental cost of retraining a DINOv2 ViT-g model, as well as the cost of the whole project. I find the latter point particularly interesting. The authors write they estimate the cost of the whole project between 0.5k and 1k tCO2eq. Maybe providing some context for this number would be helpful for a reader to have a better understanding of this number. For comparison, a seat on one return flight between London and New York emits around 1.8 tonnes of carbon dioxide (tCO2) (co2.myclimate.org). This means, this project cost an equivalent of between 278 and 556 single person return flights. A Boeing 777 carries 312 passengers, so the cost of the project is equivalent to one-two return flights of a full Boeing 777 between London and New York. There are somewhere between 10 and 36 daily flights from London to NY, which probably makes the environmental impact of this project negligible overall. -> Not sure if this suggestion/ calculation is helpful, but at least for me, putting the total project cost in relation to single-person flights helped me better understand this cost. This may or may not be a helpful addition to this section of your paper.

### Weaknesses:

[major] „We release all the models and the code to retrain DINOv2 on any data.” How about the curated dataset LVD-142M? If the dataset will be released, please explain in what way. Most (if not all?) of the constituent datasets (Table 15, Appendix) are publicly available, but not in a convenient pipeline where it would be easy to build LVD-142M from scratch. Thus, releasing the curated dataset in some form would allow the research community to reproduce the presented results. Further, the released foundation models will be used by a lot of people for a multitude of tasks. We know that dataset biases affect the trained models. Not being able to access the training data also prevents the community from assessing the biases inherited by the models. **I would like to also get input from my fellow reviewers and the Action Editor, but to me, releasing the training data is a necessary requirement to recommend accepting this paper.**

I have listed further issues I found in this paper in the field below.

---

### Review · Reviewer_6PH8 · 2023-08-24

**Summary Of Contributions:**

The paper proposes a pre-training dataset and pre-training method for a large ViT model. The paper proposes and studies the effects of different data sources/filtering, training loss components/tasks, and makes an efficient training implementation. The approach is evaluated on many standard image datasets.

**Audience:**

Yes

**Claims And Evidence:**

Yes

**Requested Changes:**

Overall the paper is good. Answering the above questions would be great.

**Strengths And Weaknesses:**

Strengths: The paper is well written, easy to understand and follow. The paper makes multiple meaningful contributions, such as efficient implementations, dataset contribution. The code and model claims to be open sourced (will the data be released too?). The experiments are thorough, show the effects of all the components and have useful insights.


Questions:

Is the deduplication done between the uncurated images and the initial training sets (e.g. imagenet) or only on the uncurated images?

The uncurated dataset is filtered to include images similar to those in the training data, e.g., imagenet. Does this harm generalization to other tasks that have data not like imagenet? For example, OCR, understanding graphs and image data of other types? Table 2 indicates that the uncurated data hurts performance on the chosen tasks, which makes sense, but does the curated version hurt generalization?

"Untying head weights between both objectives" Could this be clarified? A bit unclear exactly what is meant by this and seems significant to do.

"The distilled model outperforms the one
trained from scratch on 10 out of 12 benchmarks" But figure 5(a) shows it outperforming the ViT-L scratch on all 12?

Why does the small distilled model outperform the larger model on paris and oxford? Any insights on that result?

---

### Review · Reviewer_7CGu · 2023-08-31

**Summary Of Contributions:**

The goal of this paper is to extract pretrained image features that are useful “out-of-the-box” (without fine tuning).  The proposed approach combines a number of recent ideas from the literature and trains on a large dataset curated in a particular way detailed in the paper.

The main model is primarily based on a combination of recent iBot and Dino approaches.  The authors also describe a number of techniques for efficient training.  Ablations demonstrate the value of the curation process and each component of the model/

The final “Dino v2” model is evaluated in many ways : some highlights include
* Outperforming Open CLIP and EVACLIP on the Imagenet linear probing benchmark,
* Strong generalization to out of domain data as shown on a number of domain generalization benchmarks, and
* Outperforming other SSL and weakly supervised approaches on instance recognition, semantic segmentation and depth estimation.  On some datasets, the performance approaches that of fully supervised SOTA.

**Audience:**

Yes

**Broader Impact Concerns:**

Sufficiently addressed in paper.

**Claims And Evidence:**

Yes

**Requested Changes:**

Some questions/suggestions to the authors below.  My vote for acceptance is not conditioned on addressing these points.

* I’m curious to understand why the gap to competitors is so big on instance recognition?  What about this problem is hard for the other approaches that Dino v2 solves?  And what is absolute SOTA on this task?
* There is no ablation of the data pipeline components but this would also be valuable to know — for example, how much of an impact does the de-duplication part of the pipeline have on the final results?
* In a few places in the paper there are claims about improving the stability of training, but this doesn’t ever seem to really be quantified in any way.
* I’m somewhat confused about the curation process for the dataset — is the reason why we want to sample from the uncurated set to boost the representation of the underrepresented parts of the curated set?
* I did not understand the Nested Tensors paragraph
* For completeness I would recommend summarizing the iBot losses in more detail since this is such an important part of the approach taken in this paper

Small nits:
* Misspellings:
** Page 1, “Additionnally”
** Page 11, “litterature”
* Data sources section on page 4: can you say what “publicly available repository of crawled web data” was used?
* Table 1: what is the “128k prototypes” line?
* Fig 4 axis labels are cut off (possibly from overaggressive vspacing)
* Table 4: there are different versions of the LAION dataset — can you say which one was used here?
* Robustness analysis section on page 11: add citations for imagenet A,R, sketch
* Semantic segmentation section on page 14: the third paragraph of this section wants to have a bold header for readability.  This sounds like a very nice result - can you add more specifics here?  What fraction of weights is fine tuned? how much time does it take?
* Class labels in the qualitative examples are quite difficult to read.

**Strengths And Weaknesses:**

Overall this is a well polished paper containing an exhaustive set of evaluations.  It’s probably better to view this as a systems paper rather than an algorithmic contribution as there is limited technical novelty.  This being said, I found many of the implementation details in the paper quite interesting and the results are very strong.

The major contribution of this paper will be a set of high quality and versatile features that are publicly available and usable out-of-the-box, with good out-of-distribution generalization, beats OpenCLIP on many tasks, that are also very good for instance recognition and dense problems in general. This will be useful to the community and based on these strengths, I recommend acceptance.

---

### Author Response · Authors · 2023-09-11
**Added revision following the reviewers' remarks.**

This is just a comment to point out we updated the submission with a new revision following reviewers' remarks.

---

### Decision · Action_Editor_KibY · 2023-12-06

**Recommendation:** Accept as is

**Comment:**

As acknowledged by all reviewers, it is a solid, high quality paper that should be accepted at TMLR. The main outcome of the paper is a foundation model accompanied by the training dataset (with the code for curation) that the authors have promised to release. Both pretrained features and the dataset are expected to be of high interest to the community.

**Audience:**

The paper will find a strong interest from the TMLR audience since both the pretrained foundational model and the training code is being released by authors. Authors have also promised to release the dataset and the code for the data curation pipeline, which would potentially be of great interest to the community.

**Claims And Evidence:**

The paper presents DINOv2, a foundational vision transformer model trained in a self-supervised manner on a new dataset LVD-142M curated by the authors. As pointed out by the reviewers, the primary contribution of the paper is a set of pretrained features that are usable out-of-the-box, with good out-of-distribution generalization, and are competitive or better than several existing self-supervised and weakly-supervised methods.

As the authors claim in the paper, the main technical contributions are in successfully scaling the training in terms of model and data size, while the technical/algorithmic ingredients in the paper (losses, regularizers, layerscale and stochastic depth for training stability) are inspired from recent papers in the literature.

All reviewers agree that the authors have presented strong empirical support for the effectiveness of the trained features on a variety of vision tasks, including on out-of-distribution tasks.